# Analysis of OpenStreetMap Data Quality for Selected Counties in Poland in Terms of Sustainable Development

Sylwia Borkowska *[ID] and Krzysztof Pokonieczny [ID]

Institute of Geospatial Engineering and Geodesy, Faculty of Civil Engineering and Geodesy, Military University of Technology (WAT), 00-908 Warsaw, Poland; krzysztof.pokonieczny@wat.edu.pl
* Correspondence: sylwia.borkowska@wat.edu.pl

**Abstract:** One potential source of geospatial open data for monitoring sustainable development goals (SDG) indicators is OpenStreetMap (OSM). The purpose of this paper is to provide a comprehensive evaluation of the spatial data quality elements of OSM against the national official data—the database of topographic objects at a scale of 1:10,000. Such spatial data quality elements as location accuracy, data completeness and attribute compatibility were analysed. In the conducted OpenStreetMap tests, basic land-cover classes such as roads, railroads, river network, buildings, surface waters and forests were analysed. The test area of the study consisted of five counties in Poland, which differ in terms of location, relief, surface area and degree of urbanization. The best results of the quality of OSM spatial data were obtained for highly urbanized areas with developed infrastructure and a high degree of affluence. The highest degree of completeness of OSM linear and area objects in the studied counties was acquired in Piaseczyński County (82%). The lowest degree of completeness of the line and area objects of OSM in the studied counties was obtained in the Ostrowski County (51%). The calculated correlation coefficient between the quality of OSM data and the income per capita in the county was 0.96. The study complements the previous research results in the field of quantitative analysis of the quality of OSM data, and the obtained results confirm their dependence on the geometric type of the analysed objects and characteristics of test areas, i.e., in this case counties in Poland. The obtained results of OSM data quality analysis indicate that OSM data may provide strong support for other spatial data, including official and state data. OSM stores significant amounts of geospatial data with relatively high data quality that can be a valuable source for monitoring some SDG indicators.

**Keywords:** data quality; OpenStreetMap; open data; VGI; sustainability

## 1. Introduction

The issue of spatial data quality has been attracting broad interest for many years, not only of data producers and distributors, but also from users and researchers. The importance of data quality in business and science is well recognized and widely described [1–4]. From the perspective of a data provider or distributor, quality assessment is one of the key elements of production that is always analysed in the context of compliance with technical specifications. The ability to create, collect, store, maintain, transmit, process and present information and data to support business processes in a timely and cost-effective manner requires both an understanding of the characteristics of information and data that determine its quality and the ability to measure, manage and report on it. The issue of geospatial data quality has a crucial role in terms of its ability to be used to determine sustainable development goals indicators. In 2015, the United Nations adopted 17 sustainable development goals (SDGs), which represented a universal call to action to end poverty, protect the planet and ensure peace and prosperity for all by 2030 [5]. To track progress toward sustainable development goals, the UN proposed a set of 231 statistical indicators that range from health outcomes such as infant mortality (Indicator 3.2) to economic indicators such as the percentage of the population living in poverty (Indicator 1.1), environmental indicators

such as air quality (Indicator 11.6) and geospatial data. Indicator-based approaches help ground broad and, in many cases, vague sustainable development goals in more concrete and measurable terms, but obtaining the data needed to monitor indicators on a national or global scale is a significant and fundamental challenge.

Data quality problems are quite widely highlighted by the international standards organization ISO (the International Organization for Standardization). ISO has developed several standards dedicated to the assessment and reporting of geospatial data quality, including: ISO 8000-61:2016—data quality and ISO 19157:2013—geographic information—data quality. ISO 8000-61:2016 specifies the processes required for data quality management. Each process is defined by a purpose, outcomes and the activities that are to be applied for the assurance of data quality [6].

The following are within the scope of this part of ISO 8000: fundamental principles of data quality management, the structure of the data quality management process, definitions of the lower-level processes for data quality management, the relationship between data quality management and data governance and implementation requirements.

The scope of this part of ISO 8000 does not include detailed methods or procedures by which to achieve the outcomes of the defined processes.

ISO 19157:2013 establishes the principles for describing the quality of geographic data. It defines the components for describing data quality, specifies the components and content structure of a register for data quality measures, describes general procedures for evaluating the quality of geographic data and establishes the principles for reporting data quality. ISO 19157:2013 also defines a set of data quality measures for use in evaluating and reporting data quality. It is applicable to data producers who provide quality information to describe and assess how well a data set conforms to its product specification and to data users attempting to determine whether or not specific geographic data are of sufficient quality for their particular application. ISO 19157:2013 does not attempt to define minimum acceptable levels of quality for geographic data.

In ISO normative documents, quality is defined as a comprehensive set of characteristics and features of data sets and services that affect the ability to satisfy current and future user requirements [7]. The characteristics and features mentioned in the standard with respect to spatial data sets are defined by more than a dozen quantitative and qualitative indicators. The most commonly used ones, which are also valid for INSPIRE Directive (Infrastructure for Spatial Information in Europe) data sets, include completeness (lack and excess of objects), logical consistency (conceptual, topological, domain and format), position accuracy, temporal and thematic accuracy (e.g., correctness of classification or correctness of quality attributes) and lineage [8]. All the mentioned quality elements are assessed in terms of compliance with the technical specifications of the data, and the results of the assessment are reported in the metadata. Researchers also pay attention to data availability, which is often a key element of quality, and to the distributor that guarantees better quality, e.g., official data are considered more reliable [3].

As far as data collected voluntarily and free of charge by a very large number of volunteers, referred to as volunteered geographic information (VGI) or crowdsourcing geodata are concerned, the use of the indicators mentioned above becomes problematic. This results from the lack of detailed technical specifications, only giving rules and guidelines for data provision, and the frequent absence of formal verification of all data entered into the database. Volunteers are usually left with a great deal of discretion regarding the accuracy of the data entered and the detail of its descriptive characteristics. Data verification is generally performed by other users, who are potentially more familiar with the area or willing to use community data for specific tasks. The dataset that is the most commonly studied and evaluated for data quality is OpenStreetMap (OSM), which is also a potential source of geospatial data for monitoring SDG indicators [9,10].

## 1.1. Related Works

The usage of OpenStreetMap has rapidly increased since it was first established in 2004. In line with this increased usage, a number of studies have been conducted to analyse the accuracy and quality of OSM data, but many of them focus mainly on the completeness and accuracy of the location of roads or buildings. In the study [11], the author aimed to analyse the quality of OSM data by comparing it with the Ordnance Survey (OS) data sets in England and selected five areas of London. The geometric accuracy and completeness of road sections were analysed. The analysis shows that OSM information can be quite accurate: on average at a distance of about 6 m from the position recorded by the operating system and with about 80% overlap of highway features between the two data sets. In paper [12], OSM road data was analysed to characterize the behaviour of OSM participants. The study area, Ankara, the capital of Turkey, was evaluated using several network analysis methods such as completeness, degree centrality, proximity, PageRank and a proposed method to measure contributor activation in a limited area from 2007–2017. The results show that the experience level of contributors determines the type of contribution. In general, more experience means more detailed contributions. The paper [13] analyses the spatial pattern, evolution, density and diversity of OSM road networks in Iran between 2008 and 2016 and looks to find casual relations between OSM and census statistics. This is due to the fact that OSM completeness reflects the importance of OSM data in human life. The results show that the road network in Iran considerably increased from 2008 to 2016, with road length increasing to 489,400 km in 2016 from 4300 km in 2008. Road density grew while road diversity and evenness declined. Mapping direction extended from big cities to medium or small-sized ones. Western counties located in mountainous regions are still not very active. The top active counties producing OSM data are mostly populated by urban citizens.

This study [14] aims to provide an analysis of the evolution, completeness and spatial patterns of OSM building data in China from 2012 to 2017 using two quality indicators, OSM building count and OSM building density. The development of OSM numbers from 2012 to 2017 is analysed by province in a regular 1 km$^2$ grid. The obtained results showed that the number of OSM buildings increased nearly 20 times from 2012 to 2017, and in most cases, economic (gross domestic product) and OSM road length are two factors that can influence the development of OSM building data in China. Most grid cells in urban areas have no building data, but two typical patterns (dispersion and aggregation) of high-density grid cells are among the prefecture-level divisions. In the article [15], the authors describe the methods of completeness analysis of OSM buildings and their application to various test areas in Germany. The results show that unit-based completeness measurements (e.g., total number or area of buildings) are very sensitive to modelling discrepancies between official data and OSM. The November 2011 analysis in Germany showed a completeness of 25% in the Länder of North Rhine-Westphalia and 15% in Saxony. While further analyses from 2012 confirm that the completeness of the data in Saxony increased to 23%, the pace of new data entry decreased in 2012. In study [16], the quality of OSM land use and land-cover (LULC) data is investigated for an area in southern Germany for two spatial data quality elements: thematic accuracy and completeness. The results show a substantial agreement between OSM and the authoritative dataset. Nonetheless, for this study region, there were clear variations between the LULC classes. Forest covers a large area and shows both a high OSM completeness (97.6%) and correctness (95.1%). In contrast, farmland also covers a large area, but for this class OSM shows a low completeness value (45.9%) due to unmapped areas. Additionally, the results indicate that a high population density, as present in urbanized areas, seems to denote a higher strength of agreement between OSM and the DLM (digital landscape model).

These studies show that OSM information can be quite accurate, but its value depends on the areas for which it was acquired. The aforementioned studies clearly show that the best spatial data quality results were achieved for urban areas and those of interest to OSM

users. Based on the statistical results of the research, the authors have inferred that the experience levels of contributors determine the contribution type and level of object detail.

*1.2. Research Purpose*

Taking into account the above facts, this paper presents a comprehensive assessment of the quality of OpenStreetMap volunteer data, paying attention to the aspect of imperfect semantic findings and quality assumptions. The research question posed was to determine, first of all, the completeness, location accuracy and attribute compatibility of the main land-cover classes of OSM objects in relation to the national official data collected in the database of topographic objects, which was the reference base in the conducted research. The analyses were performed for five selected counties in Poland, taking into account their diversity in terms of terrain and urbanization level, which allows them to be treated as representative samples. The data to which OSM was referred were official Polish data from BDOT10k (National Database of Topographic Objects). This study complements the previous research results in the field of quantitative and qualitative analysis of OSM data, especially in relation to the Polish territory, taking into account the diversity of land cover and development of the test areas. A novelty in the study is its comprehensive approach to assessing the quality of OSM data for the main classes of land cover in the area of a particular county. Another novelty is the analysis of these elements in relation to the indicators of sustainable development, including economic resources.

## 2. General Principles of OpenStreetMap

Due to the lack of open access to spatial data in the UK, in 2004 Steve Coast initiated the free, open and editable OpenStreetMap (OSM) project. The main mission of this project is to provide both finished maps and raw geodata to any user by volunteers around the world. Despite its name, OpenStreetMap is not just a map of roads. Roads account for 23.9% of all objects, and buildings are the most represented, accounting for 58.9% of all objects collected in the database [17]. OSM is based on the idea of an open social network and uses wiki technology, which in practice means that anyone can enter or edit any object in the database at any time. In addition, the database stores the history of edits to each object, so the effects of a mistaken vectorization or deliberate vandalism may be retracted.

*2.1. OSM Data Structure*

OSM has its own infrastructure for storing, sharing, searching and visualizing data that is not compliant with OGC (Open Geospatial Consortium) standards. OSM follows the peer production model that created Wikipedia; its aim is to create a set of map data that is free to use, editable and licensed under new copyright schemes [18]. OSM data are stored in a PostgreSQL relational database, according to the WGS84 datum (World Geodetic System 1984). Basic geometric types are used to represent the geometry, which, when combined with any labelling scheme, allow virtually any geographic object to be described. Elements are the basic components of OpenStreetMap's conceptual data model of the physical world. Elements are of three types (Figure 1):

- Nodes—represent a specific point on the earth's surface defined by its latitude and longitude. Nodes can be used to define standalone point features. For example, a node could represent a park bench or a water well.
- Ways—used to represent linear features such as rivers and roads. Ways can also represent the boundaries of areas (solid polygons) such as buildings or forests. In this case, a way's first and last node is the same. This is called a "closed way".
- Relations—multi-purpose data structure that documents the relationship between two or more data elements (nodes, ways and/or other relations).

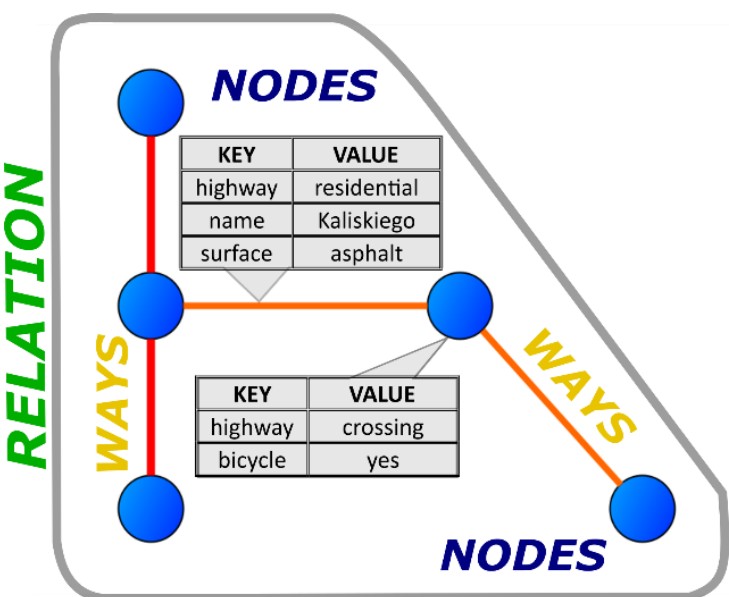

**Figure 1.** An example of an OSM object data structure.

A label, also called a tag, consists of a pair of expressions: "key = value" which can be equated with an attribute. For example, highway = residential defines the way as a road generally used for local traffic within settlement. The key highway = * is the main key used for identifying any kind of road, street or path. The value of the key helps indicate the importance of the highway within the road network as a whole (Figure 1).

### 2.2. Methods of Obtaining OSM Data

OSM data users and providers use a variety of techniques and data sources to acquire site information, including vectorization of orthoimagery, measuring a route with a hand-held GPS receiver while walking (or even biking or driving), sketching or measuring from the level of the nearest road, importing public official data, etc. [19,20]. The dependence of geometric accuracy on data acquisition methods and techniques explains the heterogeneous geometric accuracy of the data in the OSM database. In addition, the effect of heterogeneity of object acquisition techniques and devices is compounded by the effect of interpretation. Correct interpretation of the spatial position of an object, e.g., the outline of a building, especially from satellite or aerial images, requires experience, preferably supported by appropriate training.

Goodchild, an ambassador of the idea and term VGI, expressed the opinion that well-distinguishable geographic objects are less demanding in terms of training and experience for volunteer observers placing them [21]. Treating this opinion as a research hypothesis, we confirm that the location of a given spatial object varies depending on its type and the interpretive capabilities of the person editing OSM.

### 2.3. Methodological Aspects of OSM Data Quality Assessment

According to ISO 19157:2013 [7], spatial data quality is understood as a set of the following characteristics and attributes of the objects collected in a database:

- Geometric accuracy—this parameter describes the accuracy of determining the coordinates of the object. In practice, the preferred method of checking this parameter is to compare it with an independent source of higher accuracy.
- Thematic accuracy—describes the accuracy or certainty of the acquisition of an attribute value. Estimating the accuracy of a quantitative attribute is analogous to location accuracy (comparison with a more accurate data source).
- Currentness—describes the point in time or moment in time when the contents of the database match reality.

- Completeness—determines how exhaustive a set of objects is. It may refer to: excess, missing objects, their attributes or the relationships between them.
- Logical consistency—describes the consistency of the relationships recorded in the spatial database structure (conceptual, domain and topological).

The quality of OpenStreetMap data, and in particular its quantitative elements such as completeness and geometric accuracy, is of broad interest to potential users worldwide. The method of data collection used in OSM makes it impossible to directly apply the principles of geographic data evaluation contained in ISO 19157, which refer to the comparison of data with technical specifications. Goodchild [22] listed three alternative approaches to assessing the quality of geographic data acquired by projects such as OpenStreetMap:

- Crowd-sourcing approach—based on the assumption that redundant data is detected and corrected by users.
- Social approach—assuming minimal data validity checks by administrators.
- Geographical approach—involving the use of GIS-type programs for data quality control by checking the correctness of topologies and logical rules.

Such approaches to assessing the quality of VGI data are gaining popularity, although external assessment is still widely used, requiring access to other, usually more accurate and reliable data. Such an approach was used in this research because it allows for a comprehensive assessment of data quality, which is crucial from the point of view of potential users. However, the selection of reference data is a problematic issue. Concerns about what reference set to choose for quality control of spatial databases fed by non-cartographer volunteers were also expressed in studies [11,23,24]. Given their study, the official state resource was selected for comparison. Considering that most of the previous analyses of the quality and usefulness of OSM concerned large cities or agglomerations [25–27], the research results presented below are novel in that they include an analysis of spatial data covering not only roads and buildings but consider a broad data set consisting of the main land-cover elements for counties in Poland that are diverse in terms of terrain and degree of urbanization (thus, they do not cover only urbanized areas).

## 3. Materials and Methods

### 3.1. Source Data

3.1.1. Area of Research

The test area consists of five counties located in the territory of Poland. The counties that were selected are diverse in terms of location, terrain, area, degree of urbanization and land cover, so that they can be representative samples. The location of the analysed test areas is shown in Figure 2.

Piaseczyński County is situated in the central part of Masovian Voivodeship. It is one of the richest and best developed counties in Poland [28]. The county has an area of 621 km$^2$ and a population of 182,076 [29]. Within the county there are natural plant communities: meadows, peat bogs and forests. The County of Piaseczyński lies in the belt of the central Polish lowlands. The land use structure of the county is agricultural land: 49.2%, orchards: 10% and forest: 19.6%.

Sanocki County is located in the Bieszczady Mountains in southern Poland, with an area of 1159 km$^2$ and a population of 95,035. The county has an agricultural nature, with poorly developed other sectors of the economy [29]. Forestry and related wood industry are dominant here. More than 1/3 of the area is under nature protection.

Sokólski County is a county in the northeastern part of Poland with an area of 2055 km$^2$ and population of 66,686 [29]. Sokólski County has rolling and hilly hills ranging in elevation from 110 to 240 m above sea level. The county belongs to the area of the so-called "Green Lungs of Poland", i.e., ecologically developed areas with great tourist potential. In total, 24% of its area is covered by forests and forest lands. The main forest complex is the Knyszyn Primeval Forest, through which the Supraśl River flows. A significant part of the county's area is used for agricultural production. The good condition of the natural

environment and traditional farming culture in many farms contribute to the development of agriculture.

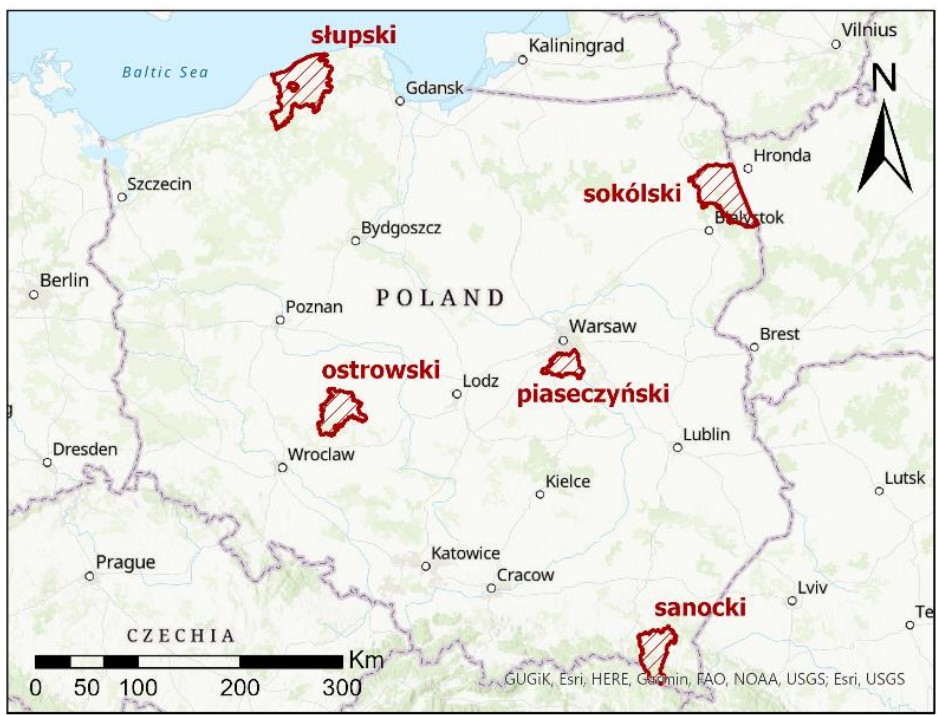

**Figure 2.** Location of analysed counties.

Słupski County is situated in the northwestern part of the Pomeranian Voivodeship, and has an area of 2304 km$^2$ (4th largest in Poland) [29]. It is bordered to the north by a 57 km long stretch of the Baltic coast. The relief is varied, with characteristic uplifts of terminal moraines and a specific, coastal landscape in the northern part, with dune areas reaching up to 30 m above sea level. Numerous rivers are an important element of the landscape—with the largest one, Słupia, being a well-known sea trout river. Protected forests cover 83% of the forest area.

Ostrowski County is a county located in the southwestern part of the Greater Poland Province and covers an area of 1160 km$^2$. The county of Ostrów Wielkopolski has the second largest population in the the Greater Poland Province (161,581 people) [29] and is situated in the macroregion of South Greater Poland Lowland. The county has an agricultural and industrial nature, with agricultural lands covering 64.9% of the county's area, and 28.3% of forest areas.

### 3.1.2. OSM Data

In the conducted OpenStreetMap data quality study, 6 main (essential) land-cover classes were used, which included linear objects (roads, railroads and river network) and area objects (buildings, surface water and forests). OpenStreetMap data were obtained from the OSM Geofabrik service [30]. The timeliness of the surveyed OSM data is 24 June 2021. Table 1 provides a description of the data used and Figure 3 shows an example visualization of the OSM data. The analysed OSM objects were selected to match the objects from the BDOT10k reference base as closely as possible.

OSM data are characterized by heterogeneous accuracy and level of detail, depending on the acquisition technique and object contour detailing, which, in turn, depend on the skill and experience of the observer. The means of obtaining OSM data are primarily measurements from handheld GPS receivers, aerial photographs and other available data sources. OSM data timeliness varies depending on volunteer activity.

**Table 1.** Characteristics of the OSM data analysed with the distinguishing "tag" [17].

| No. | Geometric Representation | Tag | Description |
|---|---|---|---|
| 1 | line | highway = {motorway, trunk, primary, secondary, tertiary, unclassified, residential} | The principal tags for the road network. They range from the most to least important. |
| 2 | line | railway = rail | Standard track for passenger or freight trains |
| 3 | line | waterway = {river, stream, tidal_channel} | rivers, streams, and other watercourses with water flowing from one place to another |
| 4 | polygon | building = * | single building outline. |
| 5 | polygon | natural = water landuse = reservoir water = reservoir | Unspecified bodies of water. Typically lakes but can also be larger rivers, harbours, etc. |
| 6 | polygon | landuse = forest natural = wood | Forest or woodland. Sometimes considered to have the restricted meaning "managed woodland or tree plantation maintained by humans to obtain forest products". |

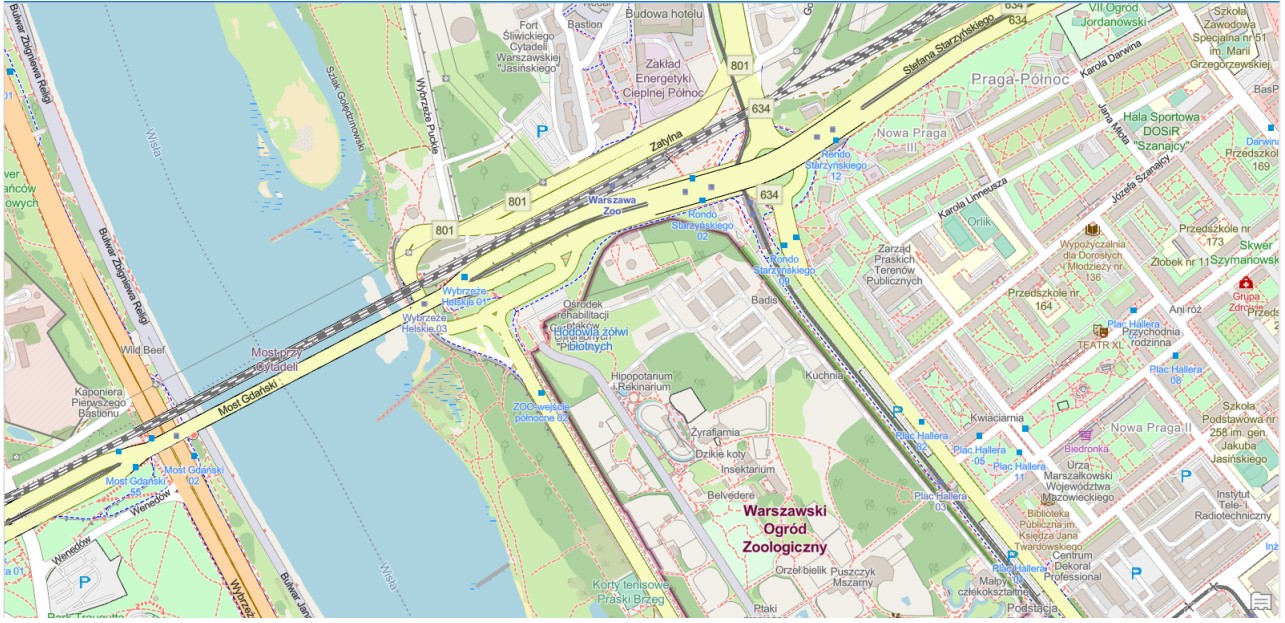

**Figure 3.** An example visualization of the OpenStreetMap database.

### 3.1.3. Reference Data—BDOT10k

In the conducted analyses of OSM data quality, the official spatial data of the National Database of Topographic Objects (BDOT10k) was adopted as the reference dataset. It is a vector database containing the spatial location of topographic objects along with their basic descriptive characteristics. The content and detail of the BDOT10k database correspond to those of a traditional topographic map at the scale of 1:10,000, (Figure 4).

The detailed scope of information collected in BDOT10k, organization, mode and technical standards of creating, updating, verifying and making data available are specified in the legal act [31]. The BDOT10k database is updated on a current basis after obtaining reliable data from feeders. The BDOT10k database is available free of charge for any use. Data may be downloaded free of charge via the GEOPORTAL service [32]. The validity of the available BDOT10k data used in this analysis is March 2020. Objects from the BDOT10k reference database, belonging to the same land-cover classes as the OSM data, were used for the OSM data quality analysis (see Table 2).

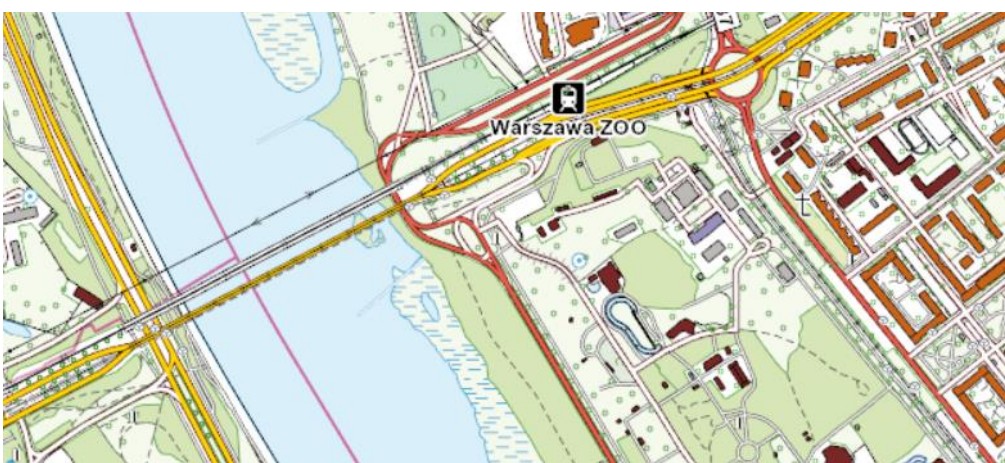

**Figure 4.** An example visualization of the BDOT10k database in Geoportal service.

**Table 2.** Characteristics of the BDOT10k data analysed [31].

| No. | Geometric Representation | Name | Designation BDOT10k | Description |
|---|---|---|---|---|
| 1 | line | road | SKJZ | Roadway centreline segments, or portions of the roadway dedicated to vehicular traffic, that have a uniform set of attributes. |
| 2 | line | track or set of tracks | SKTR | Segments of track axles or axles of sets of tracks used for the movement of railway vehicles. |
| 3 | line | river and stream/channel | SWRS/SWKN | Sections of river and stream axes between hydrographic network nodes. |
| 4 | polygon | building | BUBD | Buildings permanently connected to the ground with foundations. |
| 5 | polygon | surface water | PTWP | Areas occupied by the waters of rivers, canals and reservoirs. |
| 6 | polygon | forest or wooden area | PTLZ | Densely wooded areas: forests, wooded parks and other wooded areas. |

The BDOT10k dataset is defined in the PUWG 1992 (Państwowy Układ Współrzędnych Geodezyjnych) rectangular coordinate system, which is a coordinate system based on the Gauss–Krüger mapping for the GRS80 ellipsoid in one ten-degree zone for Poland (EPSG: 2180). The PUWG 1992 is intended for maps with a scale of 1:10,000 and smaller.

BDOT10k data are acquired through: geodetic survey, land and building registry, orthophoto vectorization or other official state registers. BDTO10K quality control is performed in accordance with the control system for data submitted to the BDOT10k resource (topology and geometry checks, semantic, syntactic and attribute checks, etc.) and is carried out in high detail.

### 3.2. Methodology

This part describes the methods used to assess the quality of OSM data for the test counties (Piaseczyński, Słupski, Ostrowski, Sanocki and Sokólski) in relation to the BDOT10k reference database.

The spatial data characteristics summarized in Section 3.1 show the basic sources of discrepancies between the analysed datasets, namely: different conceptual model, measurement rules and technological supervision and the control system of data transferred and stored in the OSM and BDOT10k databases. Taking into account the mentioned differences in both datasets, the research work included: the identification of corresponding objects in both sets as well as the analysis of geometric accuracy and completeness of objects

and attributes. Sections 3.2.1 and 3.2.2 describe the procedures for assessing the quality of OSM data in terms of geometric accuracy. Due to the different geometric type of the analysed land-cover objects, OSM data quality assessment for area (Section 3.2.1) and linear (Section 3.2.2) objects is detailed. Sections 3.2.3 and 3.2.4 describe the method for assessing the completeness of OSM data, and Section 3.2.5 analyses semantic and attribute accuracy of OSM. These methods take into account the heterogeneous nature of OSM data (linear and area objects). The analyses described were performed using GIS software—ArcGIS Pro and Statisitica software. Due to different coordinate systems of the analysed data, the OSM database was transformed to the metric system consistent with BDOT10k—PUWG 1992 uniform for the entire territory of Poland. The use of the metric coordinate system made it possible to perform spatial analyses in GIS software in accordance with the applied methodology. The transformation between the WGS 84 geographic system and PUWG 1992 was performed in accordance with the conversion tools of ArcGIS Pro. The process of performing the transformation of the coordinate system does not affect the results obtained.

3.2.1. Analysis of the Geometric Accuracy of OSM Area Objects

In order to obtain statistical information about the geometric accuracy of OSM surface objects in comparison to the reference database BDOT10k, homology points were used. Accuracy analysis was based on automatic measurement of corresponding vertices of OSM area objects in relation to BDOT10k—Figure 5.

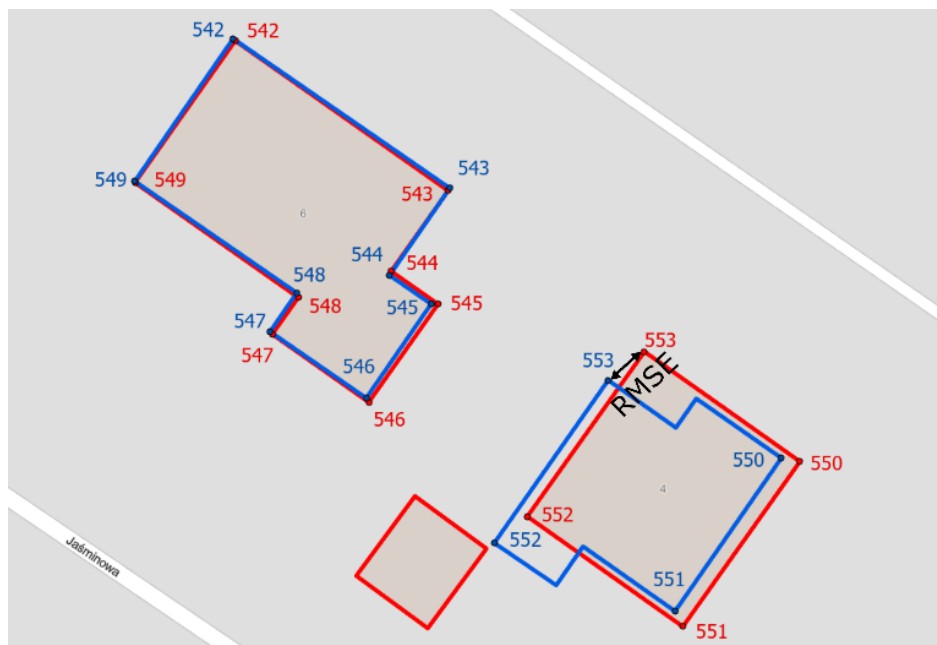

**Figure 5.** Homologous points semi-automatically detected between the OSM buildings (red) and the BDOT10k buildings (blue).

The measurement was conducted for all analysed counties, taking into account area objects—buildings, forests and surface waters. Measurement of homologous points was performed by comparing coordinates of corresponding corners (vertices) in OSM and BDOT10k databases.

The geometric accuracy of OSM area objects is presented in terms of root mean square error (RMSE). RMSE is a frequently used measure of the differences between values (sample or population values) predicted by a model or an estimator and the values observed. root mean square error (RMSE) is the standard deviation of the residuals (prediction errors). Residuals are a measure of how far from the regression line data points are. RMSE is a measure of how these residuals are spread out. In other words, it tells us how

concentrated the data is around the line of best fit. root mean square error is commonly used in climatology, forecasting and regression analysis to verify experimental results [33].

The value of RMSE error for the determined homologous points in OSM and BDOT10k datasets was calculated according to the Equations (1)–(3):

$$RMSE_X = \sqrt{\frac{\Sigma_i (X_{OSM} - X_{BDOT10k})^2}{N}} \tag{1}$$

$$RMSE_Y = \sqrt{\frac{\Sigma_i (Y_{OSM} - Y_{BDOT10k})^2}{N}} \tag{2}$$

$$RMSE = \sqrt{RMSE_X^2 + RMSE_Y^2} \tag{3}$$

where:

$X_{OSM}$, $Y_{OSM}$—coordinates of a point from the OSM database,

$X_{BDOT10k}$, $Y_{BDOT10k}$—coordinates of a point from the BDOT10k database and $N$—number of observations (homology points).

### 3.2.2. Analysis of Geometric Accuracy of OSM Linear Objects

The buffer zone method developed in research [34] was used to determine the accuracy of the location of OSM linear objects relative to the BDOT10k database. Using this method, accuracy is determined by the percentage of an OSM linear object located within the buffer zone of the corresponding linear feature from the BDOT10k reference data. The buffer zones were determined for the BDOT10k data. For each of the linear objects' class (roads, railroads and rivers), 4 buffer zones with widths of 1, 2, 5 and 10 m were delineated—Figure 6.

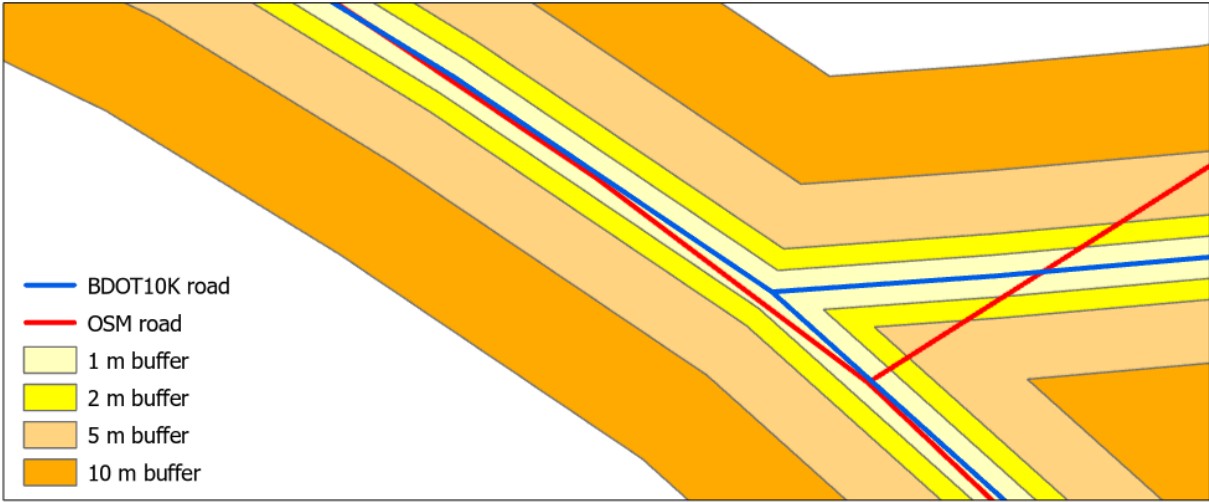

**Figure 6.** Created buffer zones around a linear object of the BDOT10k database.

The widths of the buffer zones were optimally selected on the basis of the literature and the accuracy of the BDOT10k data. The assumed accuracy of the BDOT10k database corresponds to the accuracy of maps in the scale of 1:10,000, which results in the accuracy of the location of objects in the database amounting to 1.5–5 m, depending on the type of object in accordance with [31]. Other geometric conditions of objects are also specified: the minimum distance between vertices is 2 m, and the accuracy of mapping angles is 1 degree.

The OSM data length was then overlaid on the designated BDOT10k buffer zones to calculate the percentage of overlap using the equation as follows:

$$Coverage \ [\%] = \frac{A}{B} \cdot 100\% \tag{4}$$

where:

*A*—length of the OSM dataset tested in the buffer.
*B*—total length of the tested BDOT10k dataset in the buffer.

### 3.2.3. Analysis of Completeness of OSM Area Objects

The completeness of OSM area objects is assessed using a method based on area ratio units [14], which calculates completeness (C) as the percentage ratio between the total area of an OSM object and the total area of the corresponding BDOT10k database object within a specific spatial unit (e.g., administrative or geometric). For this purpose, the area of the analysed counties was divided into sub-areas according to a regular grid of hexagons of an area equal to 1 km² each. The choice of this cell shape is suggested by [14], who argued that the hexagonal shape of the basic field has the advantage of approximating a circle, optimally covering the study area. As noted by [35], the area ratio method may introduce an overestimation of C due to the overabundance of data available in OSM relative to BDOT10k data. For this reason, further studies are recommended in which three additional indicators are calculated: true positive (TP), false positive (FP) and false negative (FN) rates. The TP indicator (Figure 7) represents the overlapping areas of surface objects between OSM and BDOT10k, i.e., the common areas between the datasets. The FP indicator represents OSM surface objects that do not exist in the BDOT10k dataset, and the FN indicator considers BDOT10k area objects that do not exist in the OSM dataset.

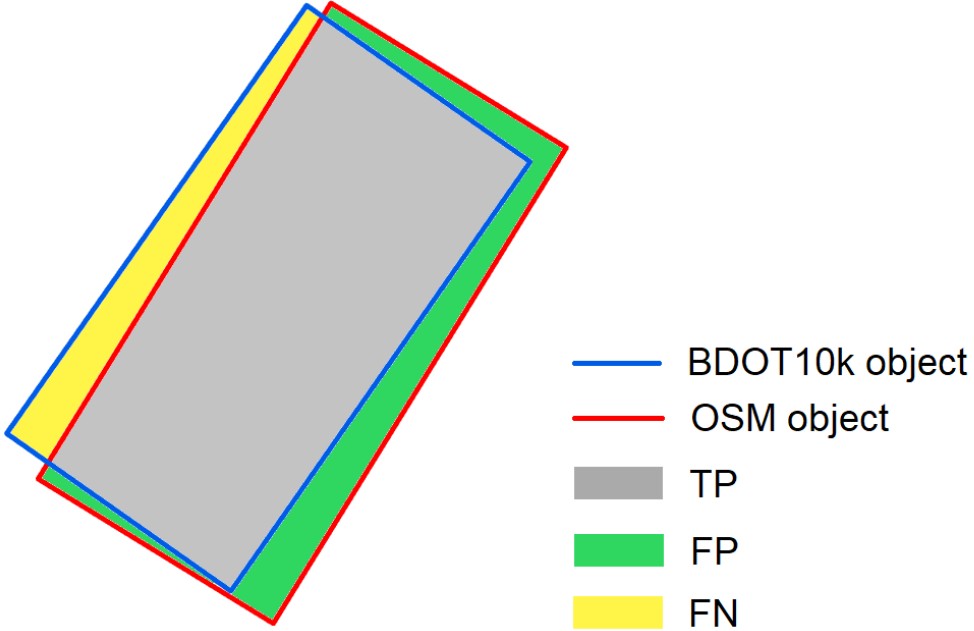

**Figure 7.** Characteristics of TP, FP and FN indicators.

To assess the completeness of OSM area objects, C, TP, FN and FP values were calculated for each hexagonal grid cell. The obtained results were then related to the total area of objects in the reference base, which is the BDOT10k collection. In the conducted research on OSM data completeness, such analyses were carried out for buildings, surface water and forests.

### 3.2.4. Analysis of Completeness of OSM Linear Objects

In this study, the completeness of roads, railroads and rivers available in the OSM database was calculated by comparing the length of a given linear feature with the length of the corresponding feature in the BDOT10k dataset (Table 3) using the Equation (5) [36]. The results are presented as percentages:

$$Completness \; [\%] = \frac{D}{E} \cdot 100\% \tag{5}$$

where:

    *D*—object length in OSM dataset.

    *E*—total length of the corresponding object according to BDOT10k dataset.

**Table 3.** Statistics on the homologous pairs detected in the considered test areas.

| County | Area Object | Number of Homologous Point Pairs | RMSE$_X$ [m] | RMSE$_Y$ [m] | RMSE [m] |
|---|---|---|---|---|---|
| Piaseczyński | buildings | 956,277 | 1.54 | 1.58 | 2.21 |
| | forests | 32,677 | 4.22 | 4.25 | 5.99 |
| | waters | 20,510 | 2.96 | 2.89 | 4.14 |
| Sokólski | buildings | 295,224 | 0.99 | 0.94 | 1.36 |
| | forests | 109,130 | 4.20 | 4.13 | 5.89 |
| | waters | 16,317 | 2.47 | 2.51 | 3.52 |
| Słupski | buildings | 288,790 | 1.14 | 1.09 | 1.58 |
| | forests | 23,625 | 3.94 | 3.90 | 5.54 |
| | waters | 15,185 | 3.35 | 3.42 | 4.79 |
| Ostrowski | buildings | 467,641 | 1.70 | 1.68 | 2.40 |
| | forests | 12,627 | 3.85 | 3.71 | 5.35 |
| | waters | 7204 | 3.01 | 2.91 | 4.19 |
| Sanocki | buildings | 285,385 | 1.43 | 1.49 | 2.06 |
| | forests | 55,834 | 3.88 | 3.87 | 5.48 |
| | waters | 7646 | 2.73 | 2.80 | 3.91 |

### 3.2.5. Attribute and Semantic Accuracy Analysis of OSM

The analysis presents quantitative results of the accuracy of OSM database attribute values. The BDOT10k database was not considered in this analysis. The quantitative analyses show the extent to which the selected OSM object tag is informed (contains information of the mapped feature). The analysis includes the number of objects with additional tags to the main tag completed, such as NAME and others describing an additional OSM object type. Attribute tests were performed by analysing the number of objects in a given OSM class and by examining the degree of user-entered information about each object's attributes. The analysis was conducted for linear and area objects. The attribute quality assessment of OSM objects was calculated by comparing the correct number of object names as an Equation (6) [36]:

$$Attribute \; accuracy \; [\%] = \frac{F}{FG} \cdot 100\% \tag{6}$$

where:

    *F*—number of objects with the informed tag in OSM dataset.

    *G*—total number of OSM objects.

    Attribute accuracy assesses the accuracy of attributes captured according to the specifications of the database.

## 4. Results

### 4.1. Geometric Accuracy of OSM Area Objects

Using the methods described in Section 3.2.1, the geometric accuracy of area objects was calculated separately for each of the five analysed counties. The obtained results of OSM area object geometric accuracy based on homologous points are presented in Table 3.

According to the results presented in Table 3, it is noticeable that the smallest values of RMSE were obtained in all the counties for building type objects. The smallest error was recorded in Sokólski County—1.36 m (295,224 homologous points). On the

other hand, the lowest value of RMSE was obtained in the Ostrowski County—2.40 m (467,641 homological points).

Objects of the forest and surface water type obtained much higher values of RMSE in the analysed areas. The lowest values on a quite similar level were obtained for objects of forest type (from 5.35 m for Ostrowski to 5.99 m for Piaseczyński County). On the other hand, the values of RMSE for surface waters were slightly more diversified—from 3.52 m for the Sokólski County to 4.79 m for the Słupski County.

### 4.2. Geometric Accuracy of OSM Linear Objects

According to Section 3.2.2, the accuracy of OSM linear objects' locations was determined by creating buffer zones in relation to BDOT10k objects with widths of 1 m, 2 m, 5 m and 10 m. Then, the percentage of overlap of OSM data in relation to each buffer zone was determined. Linear land-cover objects (roads, railroads and river network) in all analysed counties were included in the analysis. The resulting analysed linear objects by county are shown in Tables 4–8.

**Table 4.** Geometric accuracy of OSM linear objects in Piaseczyński County in relation to BDOT10k objects in buffer zones.

| Buffer Zone Width [m] | Roads | | Railway | | River Network | |
|---|---|---|---|---|---|---|
| | OSM Data Length [km] | Coverage [%] | OSM Data Length [km] | Coverage [%] | OSM Data Length [km] | Coverage [%] |
| 1 | 1786.3 | 48.5 | 52.3 | 36.3 | 79.5 | 9.9 |
| 2 | 2952.7 | 80.2 | 80.5 | 56.0 | 147.4 | 18.3 |
| 5 | 4225.1 | 114.7 | 132.3 | 92.0 | 267.4 | 33.2 |
| 10 | 4696.7 | 127.5 | 169.8 | 118.0 | 336.0 | 41.7 |

**Table 5.** Geometric accuracy of OSM linear objects in Sanocki County in relation to BDOT10k objects in buffer zones.

| Buffer Zone Width [m] | Roads | | Railway | | River Network | |
|---|---|---|---|---|---|---|
| | OSM Data Length [km] | Coverage [%] | OSM Data Length [km] | Coverage [%] | OSM Data Length [km] | Coverage [%] |
| 1 | 998.6 | 20.6 | 93.6 | 62.8 | 1092.9 | 32.5 |
| 2 | 1543.8 | 31.8 | 120.9 | 81.1 | 1547.5 | 46.0 |
| 5 | 1989.9 | 41.0 | 135.2 | 90.8 | 1805.0 | 53.7 |
| 10 | 2119.2 | 43.6 | 139.5 | 93.7 | 1926.9 | 57.3 |

According to the results presented in Tables 4–8, it can be seen that the geometric accuracy of linear objects varies significantly depending on the reference buffer width used. In case of the road network, the highest increase in the share of OSM data in the analysed buffer zones was recorded in Słupski county—from 14.3% for 1 m buffer width to 50% for 10 m buffer width. On the other hand, in the county of Piaseczyński for 1 m width of the buffer there was a significant share of the OSM data—48.5%. With increasing buffer width, the share of OSM data relative to BDOT10k increased up to 127.5% for 10 m buffer width. Increase in share of OSM data above 100% indicates data overabundance, i.e., the OSM database contains more data than BDOT10k. As far as the remaining counties (Sanocki, Sokólski and Ostrowski) are concerned, the share of OSM data in relation to BDOT10k was at a similar level with the increase in reference buffer zones, but in no case did it exceed 70%.

**Table 6.** Geometric accuracy of OSM linear objects in Sokólski County in relation to BDOT10k objects in buffer zones.

| Buffer Zone Width [m] | Roads | | Railway | | River Network | |
|---|---|---|---|---|---|---|
| | OSM Data Length [km] | Coverage [%] | OSM Data Length [km] | Coverage [%] | OSM Data Length [km] | Coverage [%] |
| 1 | 1636.5 | 23.4 | 77.3 | 44.0 | 92.1 | 15.9 |
| 2 | 2870.4 | 41.0 | 121.0 | 68.9 | 173.3 | 30.3 |
| 5 | 4338.9 | 61.9 | 147.4 | 83.9 | 317.1 | 54.8 |
| 10 | 4848.8 | 69.2 | 153.5 | 87.4 | 385.9 | 66.7 |

**Table 7.** Geometric accuracy of OSM linear objects in Słupski County in relation to BDOT10k objects in buffer zones.

| Buffer Zone Width [m] | Roads | | Railway | | River Network | |
|---|---|---|---|---|---|---|
| | OSM Data Length [km] | Coverage [%] | OSM Data Length [km] | Coverage [%] | OSM Data Length [km] | Coverage [%] |
| 1 | 1167.1 | 14.3 | 77.4 | 38.5 | 114.7 | 8.1 |
| 2 | 2022.4 | 24.8 | 117.3 | 58.5 | 216.9 | 15.2 |
| 5 | 3337.6 | 40.9 | 152.9 | 76.2 | 429.4 | 30.2 |
| 10 | 4082.0 | 50.0 | 167.4 | 83.4 | 599.4 | 42.1 |

**Table 8.** Geometric accuracy of OSM linear objects in Ostrowski County in relation to BDOT10k objects in buffer zones.

| Buffer Zone Width [m] | Roads | | Railway | | River Network | |
|---|---|---|---|---|---|---|
| | OSM Data Length [km] | Coverage [%] | OSM Data Length [km] | Coverage [%] | OSM Data Length [km] | Coverage [%] |
| 1 | 1216.3 | 24.9 | 89.0 | 35.3 | 70.1 | 6.0 |
| 2 | 1960.3 | 40.1 | 138.2 | 54.8 | 120.6 | 10.3 |
| 5 | 2733.5 | 55.9 | 238.5 | 94.6 | 196.4 | 16.8 |
| 10 | 2996.8 | 61.3 | 285.0 | 113.1 | 230.5 | 19.8 |

The analysis of the railroad network revealed that the situation is slightly different than in the case of the road network. The highest share of OSM data for the reference buffer width of 1 m was recorded in Sanocki county—62%. As the width of the buffer zones increased, the share of OSM data increased slightly—up to 93.7% for a 10 m wide reference buffer. For the Piaseczyński and Ostrowski Counties, the situation is very similar. The share of OSM data in individual buffer zones increases significantly with the increasing width of the reference buffer—from about 30% for 1 m width to over 113% for 10 m width. In the case of the Sokólski and Słupski Counties, the share of OSM data in relation to the intervals of the reference buffer zones is similar and does not exceed 90% in the case of the 10-m-wide buffer.

As for the river network, the share of OSM data in particular buffer zones is the smallest in comparison with the other analysed objects of the road and railroad network. The smallest share of OSM objects of the river network in the analysed buffers was recorded in the Ostrowski County—from 6% for the 1 m buffer to only 19.8% for the 10 m buffer. For Sanocki County, the share of OSM data in relation to BDOT10k data for the width of reference for a buffer of 1 m is the largest share from all analysed counties—32.5%. With the increase in the buffer width, this share increases to 57.3% for 10 m width of the reference buffer. On the other hand, in the Sokólski County, the share of OSM data in relation to BDOT10k reaches the highest value for the reference buffer of 10 m width—66.7%. In the

case of Piaseczyński and Słupski Counties, the obtained values of the accuracy of the OSM data position in relation to the reference buffer zones are similar and amount from about 8% for the buffer zone of 1 m to about 42% for the buffer zone of 10 m width.

### 4.3. Completeness of OSM Area Objects

The methods described in Section 3.2.3 were used to calculate the completeness of OSM area objects in individual cells of the hexagonal grid. The analysed area was divided into basic fields in the form of a hexagonal grid of 1 km$^2$. Finally, 60 thematic maps were developed visualizing the spatial distribution of C, TP, FP and FN indices in the analysed counties. Due to the large number of maps, the obtained thematic maps are presented for the two analysed counties, which represent the greatest diversity of the results. For Piaseczyński County (Figure 8) the developed maps are presented in Figure 9 and for Sokólski County (Figure 10) the obtained maps are shown in Figure 11.

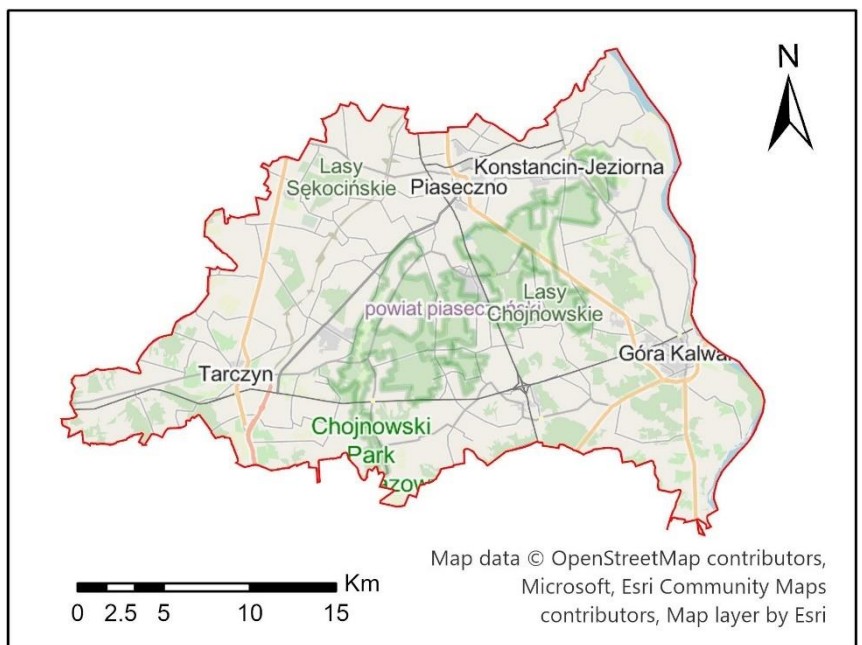

**Figure 8.** Piaseczyński County—overview map.

The set of maps presented in Figures 9 and 11 shows the spatial distribution of the calculated indices of completeness of OSM area objects in comparison with the BDOT10k reference database in two counties: Piaseczyński and Sokólski. The ranges of values for each of the indicators were determined according to the Natural Breaks algorithm. Jenks Natural Breaks classification (or optimization) is a data classification method designed to optimize the distribution of a set of values into "natural" classes. The range of classes consists of elements with similar characteristics that form a "natural" group in the data set [37].

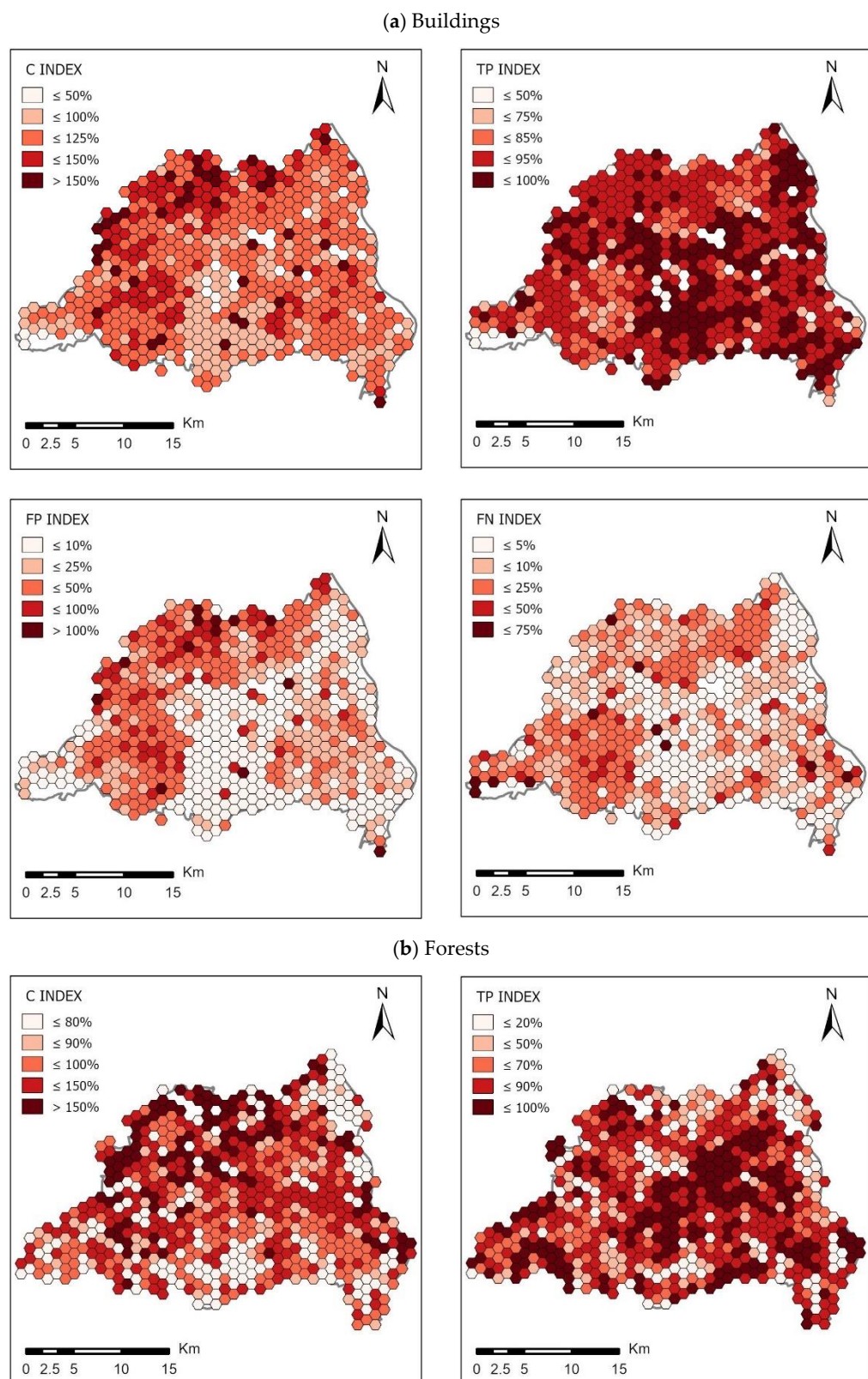

**Figure 9.** *Cont.*

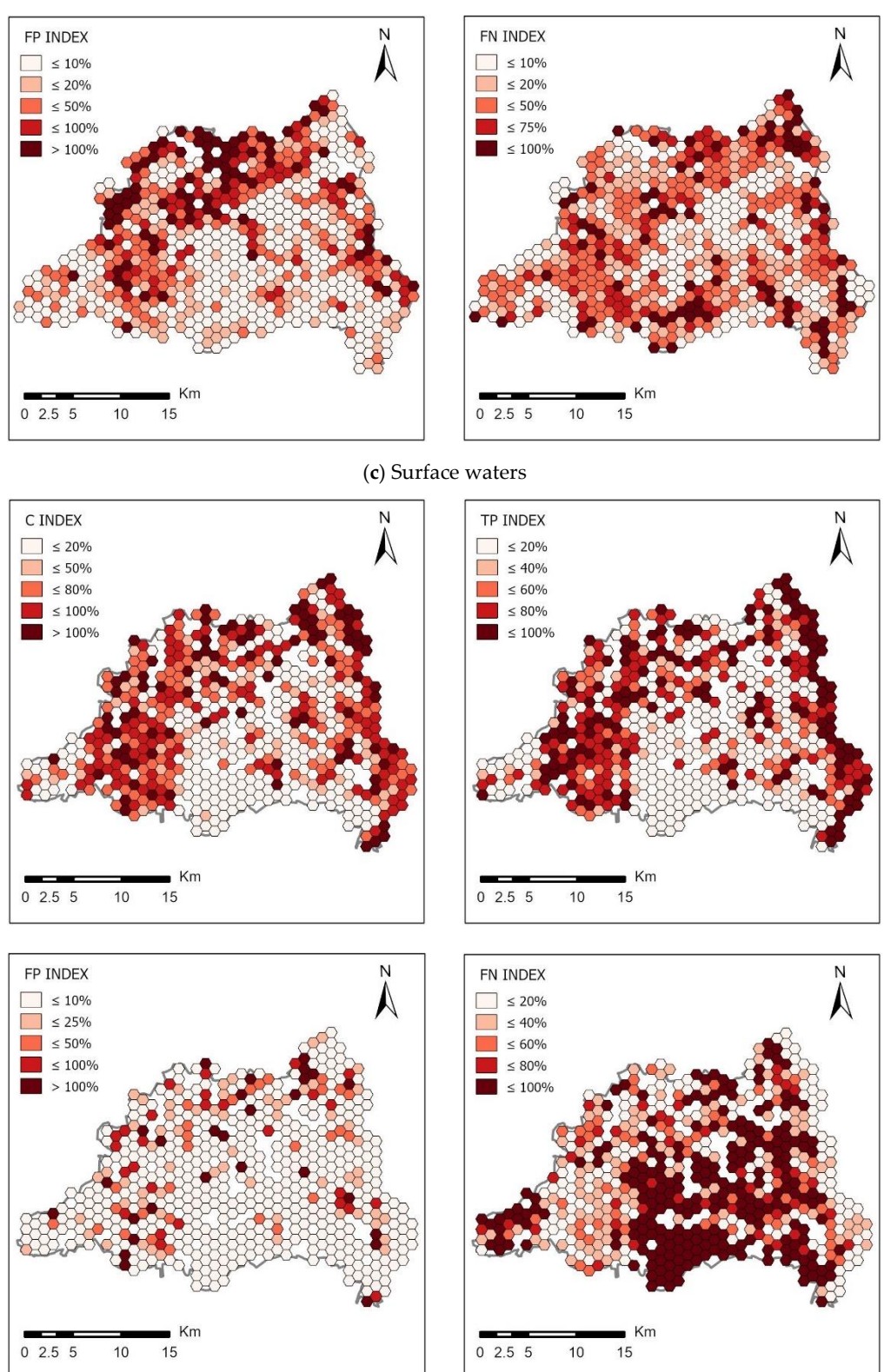

(**c**) Surface waters

**Figure 9.** Completeness analysis of OSM area objects in Piaseczyński County for indicators C, TP, FP, and FN: (**a**) buildings, (**b**) forests, (**c**) surface waters.

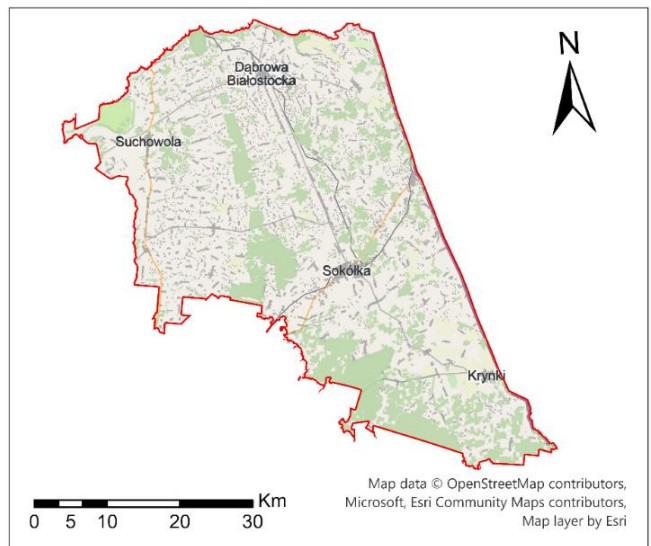

**Figure 10.** Sokólski County—an overview map.

(**a**) Buildings

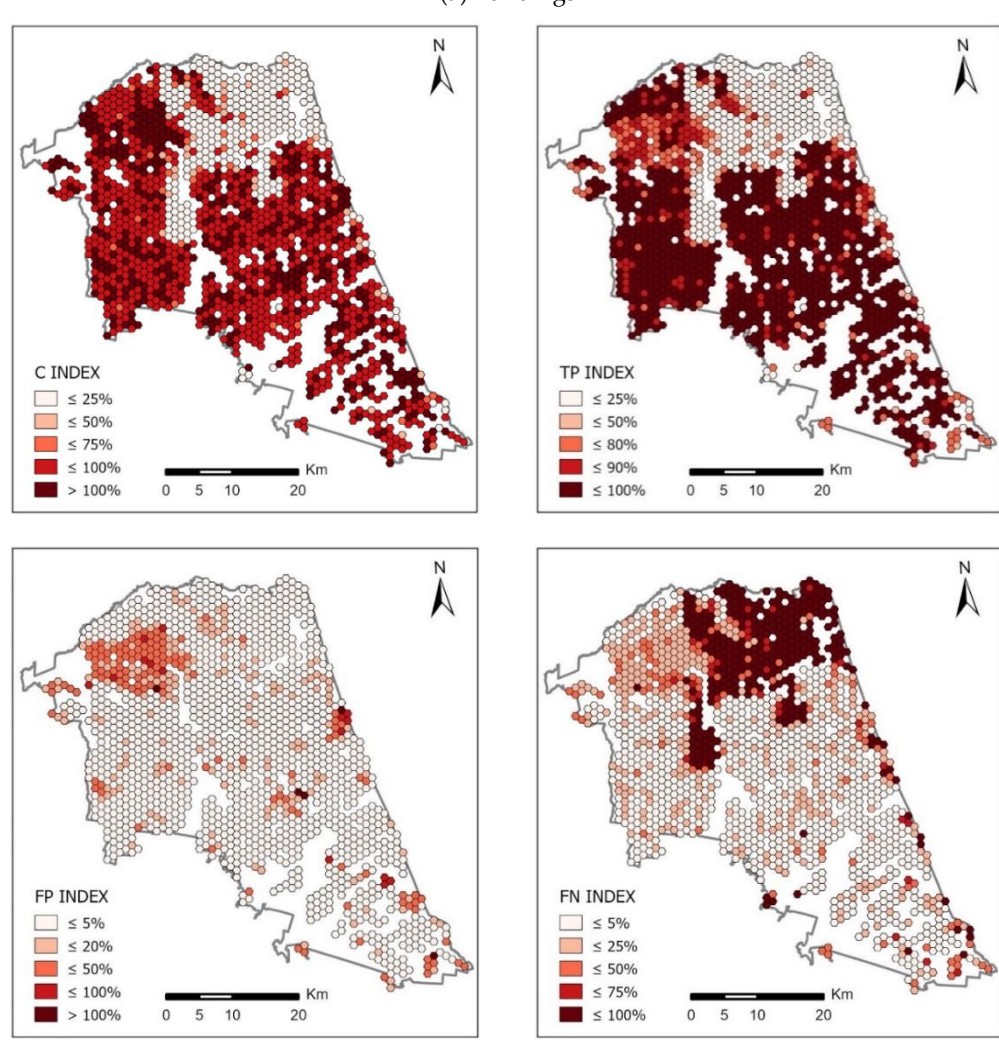

**Figure 11.** *Cont.*

(**b**) Forests

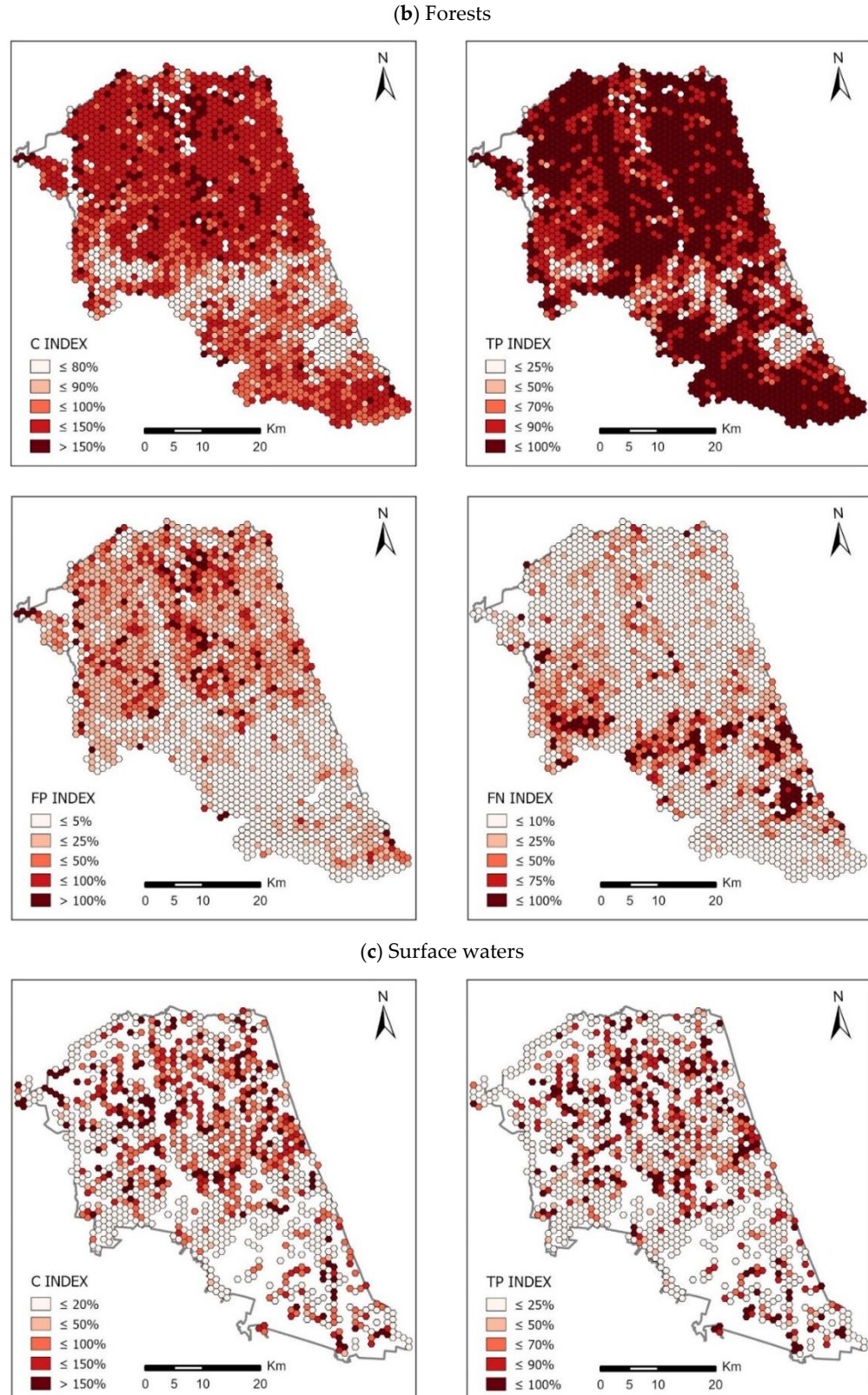

(**c**) Surface waters

**Figure 11.** *Cont*.

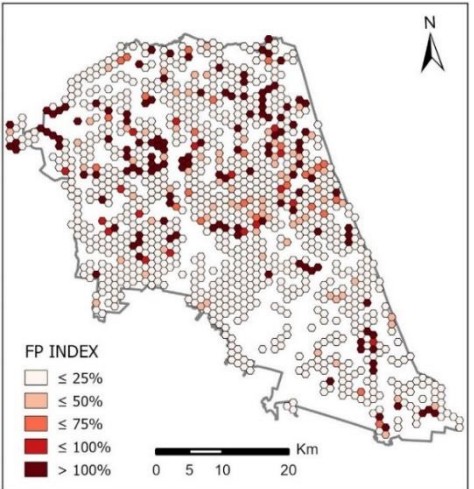
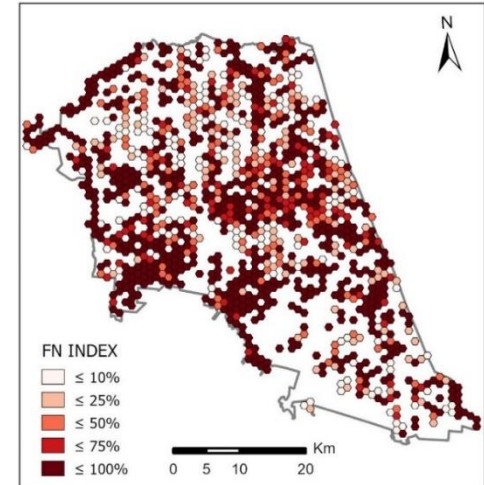

**Figure 11.** Completeness analysis of OSM area objects in Sokólski County for indicators C, TP, FP and FN: (**a**) buildings, (**b**) forests and (**c**) surface waters.

This classification method seeks to minimize the average deviation from the class mean while maximizing the deviation from the means of the other groups. This method reduces the variance within classes and maximizes the variance between classes.

The mean values of OSM area objects' completeness indices for all counties analysed with respect to the base field of the hexagonal grid are presented below in Table 9.

**Table 9.** Average values of OSM area objects' completeness indices in the analysed counties.

| Class of Area Objects | Index | Average Value of the Index in the County [%] | | | | |
|---|---|---|---|---|---|---|
| | | Piaseczyński | Sokólski | Słupski | Ostrowski | Sanocki |
| buildings | C | 115.1 | 79.5 | 66.5 | 83.5 | 111.2 |
| | TP | 89.2 | 74.3 | 52.8 | 67.9 | 92.3 |
| | FP | 25.7 | 5.3 | 14.4 | 15.6 | 18.9 |
| | FN | 10.8 | 25.4 | 47.0 | 32.8 | 7.3 |
| forests | C | 171.2 | 180.6 | 80.2 | 78.4 | 86.3 |
| | TP | 69.5 | 82.8 | 72.7 | 73.3 | 79.9 |
| | FP | 102.3 | 48.3 | 7.4 | 5.6 | 6.4 |
| | FN | 30.4 | 16.4 | 27.3 | 27.1 | 20.1 |
| waters | C | 118.8 | 154.7 | 112.6 | 35.2 | 41.7 |
| | TP | 42.7 | 32.1 | 34.6 | 25.4 | 29.6 |
| | FP | 76.2 | 122.4 | 78.3 | 10.6 | 11.2 |
| | FN | 57.3 | 67.7 | 65.7 | 75.9 | 70.3 |

The spatial distribution of the completeness index C for buildings in the analysed counties reaches the highest values in urbanized and densely built-up areas. This index in highly urbanized areas in all cases reaches values above 100%. The highest average values of index C were noted in Piaseczyński County—on average, the completeness index C here equals 115%, and in some grid cells it reaches values above 400%. The lowest value of the C indicator for buildings was observed in Słupski County—about 67%. The total area of the TP index for buildings (i.e., buildings present in both BDOT10k and OSM datasets) is approximately 89% of the total area of BDOT10k buildings on average in Piaseczyński County. The highest value of the TP index for buildings was recorded in Sanocki county (92.3%). In the Słupski County, on the other hand, this index averaged 53%—the lowest value in all the analysed counties. In all cases, the highest TP values were achieved in urban areas, where high values of index C (close to 100%) were obtained. As for the FN indicator (i.e., buildings mapped in BDOT10k but not in the OSM data set), the highest value was obtained in Słupski County—on average 47% of the total BDOT10k area, while the lowest value was obtained in Sanocki County: 7% of the total BDOT10k area. On the other hand, the total FP area (i.e., buildings mapped in OSM but not in the BDOT10k data

set) was on average 26% of the total BDOT10k area in Piaseczyński County (the highest value) and 5% in Sokólski County (the lowest).

In the case of forests, the highest C index was recorded in Sokólski County (180.6%). An equally high value of this indicator was calculated for Piaseczyński County: 171.2%, whereas the lowest C index was found in Ostrowski county: 78.4%. In all analysed counties the TP index was at a similar level, but the highest value was achieved by Sokólski County (82.8%) and the lowest by Piaseczyński County (69.5%). The FP index reached the highest value for the Piaseczyński County—102.3%. Much lower values were achieved in the three analysed counties: Słupski, Sanocki and Ostrowski (7.4%, 6.4% and 5.6%, respectively). The highest values of FN were obtained for three counties—Piaseczyński (30.4%), Słupski (27.3%) and Ostrowski (27.1%). On the other hand, Sokólski County had the lowest FN: 16.4%.

The completeness index C for surface waters obtained the highest value for the Sokólski County: 154.7%. On the other hand, the lowest value was obtained in Sanocki County (41.7%) and Ostrowski County (35.2%). The TP index obtained the highest value for Piaseczyński County (42.7%), while the lowest for Ostrów Wielkopolski County (34.6%). The value of the FP index varied significantly between the analysed counties—the highest value was achieved by Sokólski County (122.4%), while the lowest by Sanocki County (11.2%) and Ostrowski County (10.6%). The FN index for the analysed counties was at a similar level, but the highest value was calculated for Ostrowski County—75.9%, and the lowest for Piaseczyński County: 57.3%.

### 4.4. Completeness of OSM Linear Objects

As far as linear objects are concerned, OSM data completeness was assessed by comparing the length of OSM data to the length of corresponding objects in the BDOT10k reference database. The results are presented as a percentage in Table 10.

**Table 10.** Completeness of OSM linear objects in relation to the BDOT10k reference database for each county.

| County | Line Object Class | Length in OSM Base [km] | Length in BDOT10k Database [km] | Completeness [%] |
|---|---|---|---|---|
| Piaseczyński | roads | 5730.3 | 3683.1 | 155.6 |
| | railways | 177.3 | 143.9 | 123.2 |
| | rivers | 461.5 | 806.4 | 57.2 |
| Sokólski | roads | 5872.9 | 7008.0 | 83.8 |
| | railways | 157.1 | 175.6 | 89.5 |
| | rivers | 578.2 | 764.3 | 75.7 |
| Słupski | roads | 5030.0 | 8625.9 | 58.3 |
| | railways | 179.0 | 200.7 | 89.2 |
| | rivers | 951.062 | 1423.7 | 66.8 |
| Ostrowski | roads | 3525.1 | 4886.3 | 72.1 |
| | railways | 292.0 | 252.0 | 115.9 |
| | rivers | 286.8 | 1166.2 | 24.6 |
| Sanocki | roads | 2421.7 | 4857.2 | 49.9 |
| | railways | 144.9 | 148.9 | 97.3 |
| | rivers | 2147.2 | 3362.4 | 63.9 |

According to the results presented in Table 10, it can be seen that the completeness of OSM linear objects varies considerably by objects type and by the county analysed. As far as roads are concerned, the highest completeness rate was found in Piaseczyński County (155.6%).

The same applies to railroad OSM facilities, for which the highest completeness rate was again found in Piaseczyński County: 123.2%. On the other hand, the lowest index was noted in Słupski County (89.2%). The remaining counties achieved values ranging from 89.5% to 115.9%.

The highest completeness rate for river-type OSM objects was obtained for Sokólski County: 75.7%. On the other hand, the drastically lowest value was obtained for Ostrowski County—only 24.6%. For the remaining counties the obtained index of completeness was quite similar—from 57.2% to 66.8%.

*4.5. Semantic and Attribute Accuracy in OSM Database*

Tests of attribute accuracy of OSM were performed by analysing the number of objects in OSM database and by checking the degree of information entered by the user concerning particular attributes of a given object. The analysis was performed for linear and area objects in all counties. The attribute to be verified was the NAME key. Additionally, for buildings, the TYPE attribute was also evaluated, denoting information about the type of a given building. All values for buildings other than "yes", indicating a specific building type and entered values for the amenity key, were included in the analysis (see Table 11).

**Table 11.** Analysis of the attribute accuracy of OSM data.

| County | Object Class | Objects | Fields | Informed | Non-Informed | Ratio % |
|--------|--------------|---------|--------|----------|--------------|---------|
| Piaseczyński | roads | 28,924 | NAME | 11,373 | 17,551 | 39.1 |
| | rivers | 2495 | NAME | 429 | 2066 | 17.0 |
| | railways | 419 | NAME | 64 | 355 | 15.0 |
| | buildings | 148,165 | NAME | 1106 | 147,059 | 0.7 |
| | | | TYPE | 15,849 | 132,316 | 10.7 |
| | waters | 2766 | NAME | 39 | 2727 | 1.4 |
| | forests | 7716 | NAME | 95 | 7621 | 1.3 |
| Sokólski | roads | 9056 | NAME | 794 | 8262 | 8.7 |
| | rivers | 723 | NAME | 209 | 514 | 29.0 |
| | railways | 168 | NAME | 0 | 160 | 0 |
| | buildings | 49,197 | NAME | 211 | 48,986 | 0.4 |
| | | | TYPE | 938 | 48,259 | 2.0 |
| | waters | 1460 | NAME | 1 | 1459 | 0 |
| | forests | 11,631 | NAME | 15 | 11,616 | 0.1 |
| Słupski | roads | 10,575 | NAME | 2906 | 7669 | 27.5 |
| | rivers | 1209 | NAME | 388 | 821 | 32.0 |
| | railways | 179 | NAME | 0 | 179 | 0 |
| | buildings | 40,306 | NAME | 495 | 39,811 | 1.2 |
| | | | TYPE | 11,482 | 28,824 | 28 |
| | waters | 1078 | NAME | 54 | 1024 | 5.0 |
| | forests | 1556 | NAME | 9 | 1547 | 0 |
| Ostrowski | roads | 8829 | NAME | 3018 | 5811 | 34.2 |
| | rivers | 281 | NAME | 165 | 116 | 58.7 |
| | railways | 473 | NAME | 0 | 473 | 0 |
| | buildings | 76,572 | NAME | 522 | 76,050 | 0.7 |
| | | | TYPE | 20,488 | 56,084 | 26.8 |
| | waters | 581 | NAME | 68 | 513 | 11.7 |
| | forests | 729 | NAME | 12 | 717 | 1.6 |
| Sanocki | roads | 8006 | NAME | 1542 | 6464 | 19.0 |
| | rivers | 4754 | NAME | 509 | 4245 | 11.0 |
| | railways | 307 | NAME | 30 | 277 | 10.0 |
| | buildings | 45,528 | NAME | 393 | 45,135 | 0.8 |
| | | | TYPE | 4562 | 40,966 | 10 |
| | waters | 346 | NAME | 12 | 334 | 3.5 |
| | forests | 1209 | NAME | 14 | 1195 | 1.0 |

According to the results presented in Table 11, it can be seen that the degree of attribute accuracy of OSM database is very low. The highest degree of compatibility was achieved for rivers in Ostrowski county (58.7%) for the attribute "NAME". A rate of 0% was recorded

for railroads in the Sokólski, Słupski and Ostrowski counties. As for roads, the highest degree of attribute compatibility of OSM database was achieved in Piaseczyński County (39%), and the lowest in Sokólski County (8.7%). For rivers, the relatively highest results were achieved in Ostrowski County, and the lowest ones were recorded in Sanocki County: 17%. For railroads, the highest rate was recorded in Piaseczyński County: 15%. When analysing the attribute compatibility of buildings in OSM, the information degree of two attributes "NAME" and "TYPE" was examined. For the "NAME" attribute, the highest rate was achieved in Słupski County (1.2%), and the lowest rate was achieved in Sokólski County (0.2%). The situation was similar for the "TYPE" attribute: the highest index was recorded for the Słupski county (28%), and the lowest for Sokólski county (2.0%). In the case of rivers, the highest compatibility index was indicated in Ostrowski County (11.7%), and the lowest in Sokólski County (0%). In the case of forests, the attribute compatibility index remained in all counties at a fairly similar level—about 1%, but it reached the lowest in Sokólski County (0%) and Słupski County (1%).

## 5. Discussion

In the conducted research, the geometric accuracy, completeness and semantic and attribute accuracy of OSM linear and area objects, representing the main land-cover elements, i.e., buildings, forests, surface water, transportation network and water network, were analysed.

### 5.1. Geometric Accuracy of OSM Area Objects

The analysis of the geometric accuracy of area objects revealed that the highest accuracy of location was achieved by buildings—the average error of RMSE in this group was 1.92 m. The best results were achieved for counties with a high level of urbanization. In case of the Piaseczyński administrative district the achieved accuracy of RMSE of homological points in comparison with other results was not the highest, but it should be emphasized that the number of investigated pairs of homological points in this county was exceptionally big—even three times bigger than in other counties, which might have influenced the obtained results. The lowest results of accuracy of OSM surface objects' position were achieved for forests: on average the RMSE error was 5.65 m.

### 5.2. Geometric Accuracy of OSM Linear Objects

Analysing the location of linear objects, it was noted that the highest accuracies were achieved for such objects as roads and railways. As the width of the buffer zone increased from 1 to 10 m, the share of OSM objects in the given buffer also increased significantly. The highest accuracy along with the highest share of objects was recorded in Piaseczyński County, which is characterised by a dense and developed road network, while the lowest accuracy was recorded in Sanocki County, with agricultural and forest structure, where the road network is poorly developed. Quite high accuracy was also noted for railroads: as the buffer zone increased up to 10 m in the analysed counties, the share of railroads oscillated around 100%. The lowest results were obtained for the river network: from 6% for 1 m buffer (Ostrowski County) to maximum 67% for 10 m buffer (Sokólski County).

### 5.3. Completeness of OSM Area Objects

As far as spatial data completeness analysis is concerned, the results obtained were quite diverse and depended on the type of object and analysed county. For buildings, the highest OSM data completeness values were obtained in urbanized areas of the studied counties (cities and built-up areas). The lowest completeness values were found in the suburbs of the counties and in agricultural areas. For buildings in built-up areas, over-completeness was often recorded, i.e., the number of OSM buildings significantly exceeded the number of BDOT10k buildings. Additionally, the calculated TP index, showing the degree of overlap between OSM and BDOT10k objects, reached the highest values for areas, with the degree of completeness oscillating around 100%. The FP index, which informs

about OSM surface objects that do not exist in the BDOT10k data set, also achieved the highest values for highly urbanized areas, which resulted directly from the high over-completeness of OSM data. Finally, the highest values of the FN index were achieved for areas where the degree of data completeness was the lowest. In these cells, the majority were objects that were not present in the OSM database, although they existed in the BDOT10k database.

### 5.4. Completeness of OSM Linear Objects

The analysis of the degree of completeness of OSM linear objects in relation to the BDOT10k reference database for individual counties revealed that the transport network in most of the studied counties achieved the highest results, including over-completeness (numerically, the OSM base exceeds the BDOT10k base) for counties with a high degree of urbanization and a developed transport network (Piaseczyński and Ostrowski Counties). In the case of the river network, the lowest completeness index (up to 75.7%) was recorded in the Sokólski County.

### 5.5. Semantic and Attribute Accuracy in the OSM Database

The average attribute accuracy index obtained for OSM linear and surface objects was only 11.7%. The highest accuracy values were obtained for road network, rivers and buildings in developed counties that were also popular among users: Piaseczno, Ostrowski and the coastal county of Słupsk. The lowest indices were obtained for railroads, forests and surface waters. The quantitative results show that the main tag of each type of analysed OSM objects is mostly informed, while the secondary attributes are rarely informed. The obtained results indicate the need to complete information about most of the objects in the OSM database—according to the analyses performed, there is no information about the name and type of most of the OSM facilities. Lack of information value concerning buildings and roads may be a serious obstacle in using the OSM database for many spatial analyses.

### 5.6. Comparison of OSM and BDOT10k Data with Orthophotomap

Another element of the OSM data quality assessment was the comparison of objects from the OSM and BDOT10k databases with the actual terrain situation visible on orthophotomap updated to 2020 from Geoportal service. For this purpose, about 20 objects of the analysed geometric types presented on the orthophotomap were selected in random places of each county. These objects were then compared with corresponding objects in the OSM and BDOT10k databases. An example of the objects identified in the OSM and BDOT10k databases on the background of an orthophotomap is presented in Figure 12 below.

As a result of the analysis, it was found that the greatest differences were noted for buildings (outline shift in relation to the analysed databases) and forests (defining the contour of the forest was subject to the interpretation skills of the OSM and BDOT10k database editors). Additionally, it was found that the large over-completion of OSM database objects is mainly due to the entry of a given building in the OSM database and its absence in the BDOT10k database, which is updated relatively less frequently than the OSM database. Such cases occurred in single cells of the analysed hexagonal network, which led directly to a high completeness rate. In addition, it was observed that the location and outline of OSM buildings is more generalized than BDOT10k objects, which retain considerable detail of the building shape. However, as far as forests and surface waters are concerned, the OSM database retains a higher level of contour detail than the BDOT10k database.

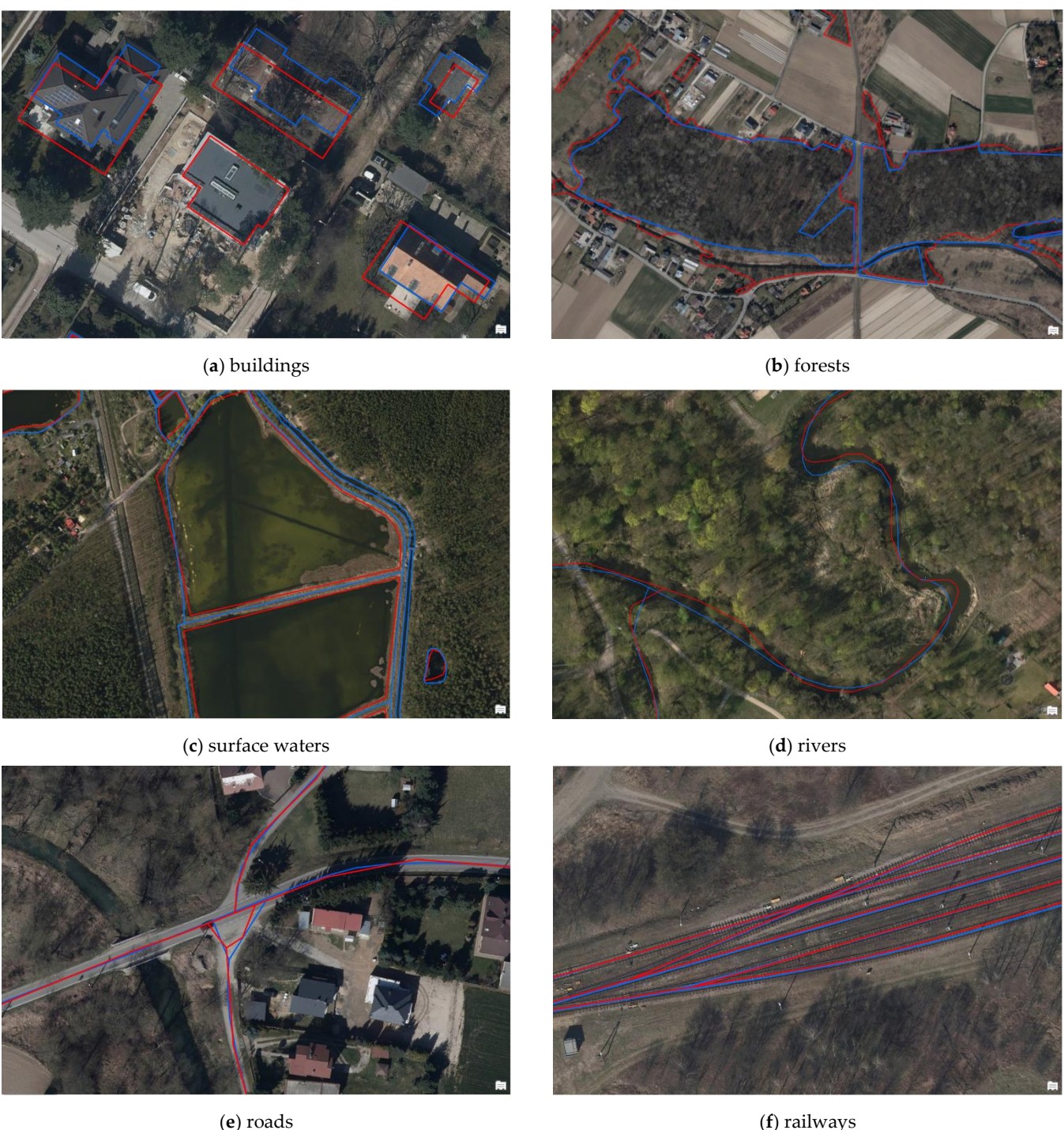

**Figure 12.** Location of the analysed OSM database objects (red) and BDOT10k (blue) on the background of the current orthophoto: (**a**) buildings, (**b**) forests, (**c**) surface waters, (**d**) rivers, (**e**) roads and (**f**) railways.

The obtained differences between the quality of OSM data in comparison with the BDOT10k reference database are certainly due to several reasons. One of them is the discrepancies in the source images and their shift in relation to reality. Many companies and institutions provide imagery to help create OpenStreetMap. In Poland, it is possible to use the data available on the official Geoportal service. They are correctly calibrated for the territory of Poland. These images are used when updating BDOT10k. The available aerial photos or satellite imagery from sources other than Geoportal are in a large proportion of

cases shifted compared with reality. In this case, the person editing OSM should suggest GPS traces. The BDOT10k database is regularly updated, while the objects in the OSM database are introduced by users on an ongoing basis. Updating and verification of BDOT10k data sets for a selected area (usually a poviat) is a long-term process subject to strict official regulations. For this reason, there are visible differences in the quality of OSM data, which make it over-completed in relation to the BDOT10k and leads to differences in the mutual spatial position (FN and FP Indices). It should also be emphasized that in the BDOT10k database, the classification of objects is strictly defined by legal regulations corresponding to the detail on a scale of 1:10,000 [31]. There are no such regulations in the assessment of OSM objects, which leads to a fairly large content for users to operate on. An example would be building mapping. In the BDOT10k database, buildings are defined as "construction objects permanently connected with the ground, having foundations separated from the space by means of building partitions (i.e., walls and covers), i.e., enclosed with walls on all sides and covered with a roof, with or without a basement with built-in house connections". In the case of the OSM base, the building is the outline of a single building created for each complex or "block" that may be associated with one single-family house or more complex buildings. In addition, the outlines can be very simplified outlines, or very closely match the shape of the building. In the case of introducing a building to OSM from satellite imagery, one should try to recognize the geometry of the building next to the ground and not the course of the roof. The OSM mapper community in Poland has a total of over 28,000 members, of which an average of more than 200 members are constantly active on a daily basis (data from December 2021) [38]. Special activity of people editing OSM is visible in the central part of Poland (Piaseczyński county) and the eastern part (Sokólski and Sanocki counties). The remaining parts of Poland, which include the Ostrowski and Słupski counties, show minimal or no activity in recent months [39]. Such a heterogeneous structure of the OSM community and its differentiation between counties affects the quality of OSM data and its level of topicality.

Some people editing the OSM database in Poland independently import objects from selected state registers (e.g., BDOT10k, the Register of Places, Streets and Addresses). According to available data these are mainly address points and outlines of buildings [40]. The highest rate of imported objects to OSM database concerns mainly the central part of Poland. For the analysed areas address points were imported, which is not included in the OSM data quality analysis. In the case of data on buildings and land-cover elements, such an import was not made in the analysed counties [40].

*5.7. OSM Data Quality in Relation to Economic Development*

From the point of view of economics, one of the main tasks of local government is the equitable distribution of public goods and the creation of conditions for socio-economic development, which promotes the implementation of the objectives of Agenda 2030. Therefore, in the next step, the average indicators of the quality of OSM data in each county were recalculated as the arithmetic mean for the obtained values and compared with the income per capita in the county in 2020 [29]. The obtained results are presented in Table 12.

**Table 12.** Average indicators of OSM data quality compared with per capita income of budgets in the analysed county.

| County | Average OSM Data Quality Index [%] | | | | Income Per Capita (PLN) |
|---|---|---|---|---|---|
| | Linear Objects | Area Objects | Attributes | Average | |
| Piaseczyński | 88.3 | 75.8 | 12.2 | 58.8 | 1638.5 |
| Ostrowski | 57.6 | 44.3 | 19.1 | 40.3 | 1268.3 |
| Sanocki | 62.5 | 47.9 | 7.9 | 39.4 | 1337.2 |
| Sokólski | 68.5 | 74.1 | 5.7 | 49.4 | 1539.2 |
| Słupski | 55.8 | 55.0 | 13.4 | 41.4 | 1381.2 |

The income of county budgets consists of: (1) own income, (2) subsidies, (3) general subvention and (4) funds for subsidizing tasks. The calculated Pearson correlation coefficient between the average indicator of data quality and income per capita in the county was 0.96. This means that the wealthier the county, the higher the indicator of OSM data quality. A graph of the analysed values is shown in Figure 13.

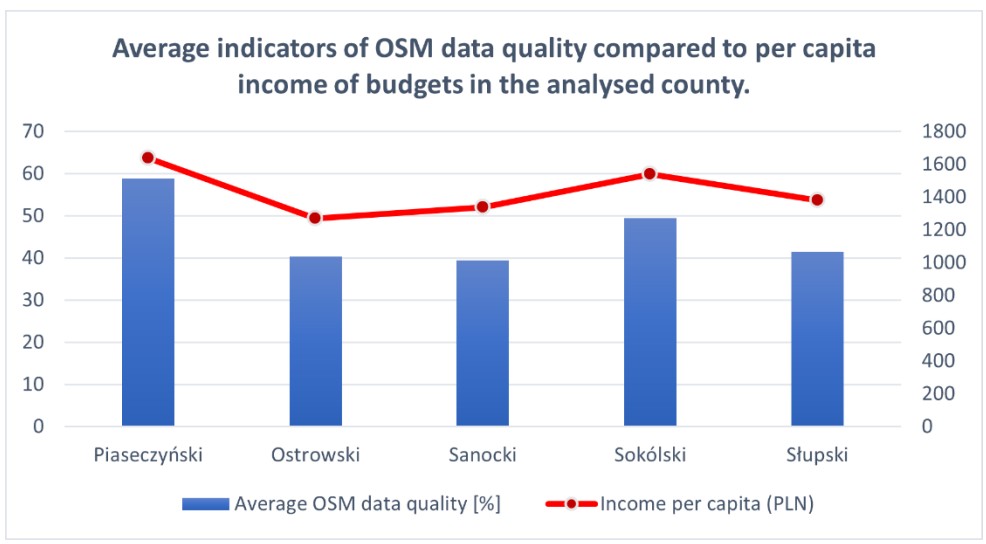

**Figure 13.** Average indicators of OSM data quality compared with per capita income of budgets in the analysed counties.

Indeed, counties with high per capita income showed the highest quality OSM data. This is probably because there are not only many more OSM objects in relatively developed regions of Poland, but also more users with high incomes and better Internet access.

OpenStreetMap can be a true data source for measuring SDG metrics that require geographic data [41]. Taking into account the demonstrated data quality, OpenStreetMap objects can be used to measure sustainable development goals, which mainly include goal 11 (make cities and human settlements inclusive, safe, resilient and sustainable) and goal 15 (protect, restore and promote sustainable use of terrestrial ecosystems, sustainably manage forests, combat desertification and halt and reverse land degradation and biodiversity loss). For example, the conducted analyses show a relatively good quality of buildings for all analysed poviats, which is taken into account by the SDG Indicator 11.7.1—the average share of built-up area in cities that is open to public use.

## 6. Conclusions

The presented results of the analysis of the comprehensive assessment of the quality of OSM data in comparison with the official reference database of spatial data BDOT10k clearly demonstrate that the obtained results largely depend on the geometric type of objects analysed and the characteristics of the test counties, including their economic development.

The obtained results confirm that the best quality indicators of OSM data were achieved for objects of quite easily recognizable and interpretable characteristics and location, i.e., buildings and transport network. On the other hand, the lowest values were achieved for objects for which defining the range and type may be a problem for a non-professional user of spatial databases—mainly forests and water network. In addition, the nature of the studied areas influenced the obtained results. The best results of spatial OSM data quality were obtained for highly urbanized areas with developed infrastructure and high per capita income ratio. The degree of coverage of line and area objects of OSM with BDOT10k amounted to 82% in the Piaseczyński county and 71.3% in the Sokólski county, respectively. The worst was in urban outskirts and low urbanized areas with low-income ratio. The obtained results show that the lowest average value of coverage of linear and

surface objects' OSM in relation to the reference database BDOT10k was obtained in the counties: Słupski 55.4%, Sanocki 55.2% and Ostrowski 51%. This is mainly due to the interest of users in a given area and the frequency of introducing new OSM spatial objects. It is also worth noting that in highly urbanized counties there was often an over-completion of data (the number of OSM data significantly exceeded the number of BDOT10k data). In less urbanized areas that are less "popular" among OSM users, there are gaps in the OSM database and "white spots" resulting from the lack of objects introduced there.

The results received from the OSM data quality analysis indicate that OSM data may provide strong support for other spatial data, including official and state data. Additionally, OSM data are mapped by users and appear in the OSM database on an ongoing basis. In the case of the BDOT10k data, the Head Office of Geodesy and Cartography, by virtue of legal provisions, conducts coordination works aimed at maintaining homogeneity, harmonization and consistency of the BDOT10k data in the entire country through cooperation with public administration bodies and voivodship marshals with respect to developing and maintaining up-to-date data. Updating the BDOT10k is a time-consuming process that involves many additional units and institutions. Due to the voluntary nature of the OSM data and the work of the database users, it should be emphasized that the database requires systematic control and supplementation with new objects and information.

Voluntary geographic information (VGI), or geospatial content generated by non-professionals who use mapping systems available on the Internet, provides opportunities for government agencies at all levels to enrich their geospatial databases. Moreover, in some cases, "eyes on the ground" VGIs have an advantage over more expensive accuracy tests conducted by official agencies because the authors have unique local knowledge. OSM's crowdsourced geospatial data helps fill micro-level data gaps and provides insight into SDG progress in a more real-time manner than is possible through annual or biennial surveys and periodic censuses. OpenStreetMap is currently the largest geospatial dataset under an open license. As OSM is increasingly used in various applications, it is important to control the quality of OSM data. OSM service provides its own OSM data quality control tools. Often the tools accomplish this by providing a list of errors in the data that mappers can then fix with editing tools. However, it is an internal tool that may validate data incorrectly. Therefore, external data quality control is important. Taking into account the received discrepancies, it would be recommended to introduce a control system in areas with available reference data of higher accuracy. Consider the OSM data quality indices presented in the article, in future studies the authors plan to extend the OSM data quality analysis with point objects for the entire territory of Poland and to compare these results with the data quality in other areas of Europe.

**Author Contributions:** Conceptualization, S.B. and K.P.; methodology, S.B. and K.P.; software, S.B.; validation, S.B. and K.P.; formal analysis, S.B.; investigation, S.B. and K.P.; writing—original draft preparation, S.B.; writing—review and editing, S.B. and K.P.; visualization, S.B.; supervision, S.B. and K.P. All authors have read and agreed to the published version of the manuscript.

**Funding:** This research was funded by the Faculty of Civil Engineering and Geodesy, Institute of Geodesy of the Military University of Technology with the frame of statutory research [PBS 933/2017].

**Institutional Review Board Statement:** Not applicable.

**Informed Consent Statement:** Not applicable.

**Data Availability Statement:** The data presented in this study is available on request from the co-author Sylwia Borkowska, e-mail: sylwia.borkowska@wat.edu.pl.

**Conflicts of Interest:** The authors declare no conflict of interest.

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
