# Peer review of "Analysis of OpenStreetMap Data Quality for Selected Counties in Poland in Terms of Sustainable Development"

_sustainability, doi:10.3390/su14073728_

Round 1
Reviewer 1 Report
Dear authors,
Thank you for the opportunity of reviewing such an interesting manuscript. scope and the case is relevant. The manuscript contains research results in the field of quantitative analysis of the quality of OSM data. The results confirm the relevance of OSM data as a potential source for monitoring indicators of Sustainable Development Goals.
The manuscript’s strengths. The general approach of the manuscript is especially good. The manuscript is informative and good structured. The title matches the content. The topic fits the Sustainability journal scope and the case is relevant. The introduction and literature review provide sufficient background and include sufficient references. Research methods were described exactly. The analysis has been performed reliably and the results have been presented in a clear manner. The conclusions match the research idea. Overall, the work deserves a high rating. The manuscript’s weaknesses.
Line 345: You should add here abbreviated title for C-completeness – (C) in order to use it later.
Line 405: I think that subsection 4.2 should be changed to 2.3.2. Please check it.
Line 486-488: You should replace the text before Table 9, Figures 9 and Figure 11.
Line 533: You should correct table numbering (please change ‘Table 12’ to ‘Table 10’).
Discussion. Authors should refer to previous research results and how they can be interpreted in perspective of future research directions; the limitations of the work should be highlighted there.
Best regards,
Reviewer
Author Response
Dear Reviewer 1,
We would like to thank you for your valuable comments and recommendations. We tried to make the revisions as carefully as we could, and we used all your comments and recommendations. Concerning the pieces of advice, we modified parts of the manuscript and marked them in red. Below, we presented, in brief, the replay to your comments.
Please see the attachment.

Reviewer 2 Report
The analysis performed is well structured and of interest in investigating OSM data quality in relation to various types of information.
A lot of numbers and indices are calculated reusing well established methods and obtaining a wide variety of elements to be discussed.
Anyway, there are a few elements that have to be highly improved to give value to all the work done.
In particular, the discussion section is not able to bring any additional consideration compared to the number that are listed in section 3. In section 3 there are al lot of numbers that need to be summarized in the discussion trying to make hypotheses on what has driven this distribution of mapped elements: e.g. whether some of the differences can be due to disalignment of source imagery, how much the fact the BDOT10k database is probably less updated has influenced some results, whether roads are or not conceptually mapped differently in urban areas or in the country, whether the two databases are considering roads in the same especially in urban areas (are footpaths and sidewalks mapped separately in BDOT10k)), ... Not everything can be analysed in a single paper, but I think that these types of considerations have to be presented and discussed in the paper.
Additionally, the differences between the completeness or accuracy are analysed without any consideration for the community of mappers and their activity: there have been any import into OSM from authoritative datasets for the investigated features? Is the Polish community well structured and homogeneuos or highly differentiated among the counties? These elements are not the core of the research, but they should be mentioned because they are an important element that influences how a territory is mapped.
One very important point is about the evaluation of the semantic accuracy. In sections 2.3.5, 3.5 and 4.5 is not clear what information has been used to compare and evaluate the semantic accuracy. Section 3.5 is mentioning NAME and TYPE information but it is no clear at all to understand what was done and what the comparison is about. So in 4.5 is not possible to understand what information OSM is missing. Please rewrite the sections explaining in details which elements of the two databases where compared and what information is missing in OSM.
Finally, both in the abstract and in the introduction there is quite an emphasis on the UN SDGs and how OSM can contribute to them, but this topic is totally missing in the core of the paper and in the discussion (only mentioned in the conclusion). If there is not specific analyses or considerations in the research about this, it should be just mentioned in the introduction, but with less emphasis in relation to the article.
Below more specific comments:
* line 9: "The purpose of this paper was" > "The purpose of this paper is"
* line 26: "not only of data producers and distributors, but also of users and researchers" > "not only from data producers and distributors, but also from users and researchers"
* line 28: "data -provider" > "data provider"
* line 73: INSPIRE is here mentioned for the first time. It could be not obvious for reader what is it. Please describe it as "INSPIRE Directive" and add a bibliographic reference.
* line 143: while I quite understand your example trying to explain the tag concept, I found it potentially misleading, because a residential road in OSM is mapped as highway=residential and there's no tag road=yes nor type=residential. Please try a different, maybe more explicit way to describe the concept, maybe with exactly the example of highway=residential, breaking down the information in a couple of steps (highway is the tag, describing a "road", while the value residential describes the "type" of road).
* line 159: even if correct, the term "OSM editor", in the OSM world, is usually used for the software allowing mappers to edit the database. So I'd suggest to rephrase "OSM editor" as "person editing OSM".
* lines 194-200: in describing the novelty of the research, the focus of the authors is about analysis of land cover, which, per se, is not a novelty in OSM quality assessment. The are already in OSM studies on data quality in relation to land use and also recently about integration of authoritative data at national level and not only at city or agglomeration (which could be cited here). The novelty seems to me to have these elements analysed also in relation to SDGs.
* line 210: "best developing counties" > "best developed counties"
* line 212: I don't know if in Polish language this is obvious, but to me it is not clear whether there's a difference between the term "Piaseczyński" and "Piaseczno" used to refer to the first described county.
* line 220: here the date of reference for the population is mentioned, while it is not the case for the other counties. The authors could probably reference at the beginning of the paragraph that the numbers regarding the population refer to this date (or a different one) for all the counties or all the counties apart Sokolski.
* line 222: "i.e. – ecologically" > "i.e. ecologically"
* table 1:
* the tag for identifying a forest is landuse=forest, there's no "forest" key. Similarly, I guess that the tag considered for water is natural=water (and no water=*))
* there are some other tags that can be considered when mapping both water and forest, but for sure one that cannot be neglected is natural=wood. Why is has not been considered? This seems to me a quite important element that may have conditioned the number for forests.
* lines 253-257: this information seems to be redundant, being already described in "2.1.2. Methods of obtaining data"
* line 282: "counties: PiaseczyÅ„ski, SÅ‚upski, Ostrowski, Sanocki, and Sokólski" > "counties (PiaseczyÅ„ski, SÅ‚upski, Ostrowski, Sanocki, and Sokólski)"
* line 293: "Subsections 2.3.3. - 2.3.4" > "Subsections 2.3.3 - 2.3.4"
* line 294: "and subsection 2.3.5." > "and subsection 2.3.5"
* line 349: "km2" > "km2"
* section 2.3.5: this section doesn't explain what information has been used to compare and evaluate the semantic accuracy. Together with section 3.5, which is mentioning NAME and TYPE information, it is no clear at all to understand what was done and what the comparison is about. Please rewrite both sections.
* line 408: "Linear land cover objects: roads, railroads and river network" > "Linear land cover objects (roads, railroads and river network)"
* table 4: it seems to me that numbers in metres are not easily readable. I suggest to use km (with 1 decimal place at maximum) instead
* table 9: the text of the items in the column "Class of area objects" is not readable for "Buildings". I suggest to keep the text horizontal.
* in figures 9 and 11 Natural Break classes are used separately for each image, but this doesn't help the comparison between the different indices nor between the 2 counties. I'd suggest to find a set of common classes that is able to represent values so they are comparable.
* lines 486: I suggest to have this paragraph moved at the beginning of section 3.3
* in whole the section 3.3, 3.4 and 3.5 I suggest to avoid using n-dashes for describing the percentages (e.g. "in Sokólski County – 180.6%") using parentheses instead (e.g. "in Sokólski County (180.6%)") or colon.
* table 10: I suggest to use integer numbers for km (there's no need to use decimals)
* all tables: to improve readability I suggest to align numbers on the right (and not centered)
* line 547: the sentence "in a given OSM database" is not clear to me. I the authors are referring to data per each county, it should be rephrased.
* lines 549-550: are NAME and TYPE fields related to the BDOT10k? This is not clear and is should be better explained the corresponding element (tag?) used for OSM. As it is, table 11 and the whole discussion about the Semantic and attribute accuracy is not clear at all.
* table 11: the name of the first county is unreadable: keep it horizontal
* lines 571-574: "of OSM linear and area objects, which are the main land cover elements, i.e. buildings, forests," > "of OSM linear and area objects representing the main land cover elements, i.e. buildings, forests,"
* line 581: "that number of investigated pairs" > "that the number of investigated pairs"
* line 582: "big – even three times bigger than in other counties," > "big—even three times bigger than in other counties—"
* line 584: "forests – on average" > "forests: on average"
* line 594: "or the river network – from 6%" > "or the river network: from 6%"
* lines 602-603: "was often recorded, i.e. the number of OSM buildings significantly exceeded the number of BDOT10k buildings)" > "was often recorded, i.e. the number of OSM buildings significantly exceeded the number of BDOT10k buildings"
* line 618: "the lowest completeness index –(up to 75.7%)" > "the lowest completeness index (up to 75.7%)"
* lines 625-626: in relation also to previous comments on sections 2.3.5 and 3.5, also in this section is not clear what are the elements missing in OSM. In the sentence "The obtained results prove the necessity to supplement the information on the majority of objects in the OSM database" what is the "information" the authors are talking about?
* lines from 630 till end of section 4.5: these are new analyses, which should go in chapter 3, not inside the discussion section.
Author Response
Dear Reviewer 2,
We would like to thank you for your valuable comments and recommendations. We tried to make the revisions as carefully as we could, and we used all your comments and recommendations. Concerning the pieces of advice, we modified parts of the manuscript and marked them in red. Below, we presented, in brief, the replay to your comments.
Please see the attachment.

Reviewer 3 Report
This manuscript describes the accuracy of Open Street Map (OSM) data for several counties in Poland. This OSM data is taken and compared with reference data whose accuracy is known and how it is acquired. However, there is room for this manuscript to be improved to further improve the quality of writing and research to be presented. Here are the comments:
- The abstract is well written. However, the authors do not highlight the results of the research findings in this abstract. Therefore, this abstract can be rephrased and improved by placing the research’s main results as described in this manuscript.
- The author stated the topological data structure in Figure 1. However, the description shown in Figure 1 is not focused on the topological data structure. It is more about how the spatial data is stored and its semantic information. The topological data structure is more specific to how each simplex in each dimension relates with other spatial objects (regardless of its space dimensions). Therefore, corrections should be made in section 2.1.1 to avoid confusing the reader.
- Since spatial data is in the form of points, lines, and polygons, there should be a description of how the data structure for polygons is formed in section 2.1.1.
- This research needs to add another type of spatial data that is spatial data based on points. In spatial science, spatial data is classified into three types: point, line, and polygon. Table 1 shows that only two spatial data types are used: line and polygon. OSM also has points spatial data on its map. But this research did not use spatial data in the form of points for the research. Therefore, it is recommended that this research include points spatial data for comparison purposes. This can further improve the quality of this manuscript.
- A description of the projection system used for the reference data - BDOT10k is required. And if the projection is different from the OSM data, how is the projection system conversion process being done? Furthermore, readers would like to know if the comparison is made using projection reference data BDOT10K or vice versa.
- For linear geometric accuracy, four buffer zones are used (1m, 2m, 5m, and 10m). Which reference is used in obtaining these four buffer zones? Which method does the author refer to in achieving this comparison? Is there any previous research using the same approach?
- As for building data, building spatial data depends on the scale and use of the application. There are applications where only buildings are presented in points only. In comparison, some buildings are presented according to the outer line of the buildings. And there are also buildings shown according to the min-max boundary only. Therefore, clarification is needed to know on what scale the building is used for OSM data and reference data so that it does not confuse the reader.
- The comparison of data accuracy with per capita income shown in the Discussion section is somewhat ambiguous. The comparison is entirely random, and further discussion is not detailed. In my opinion, it is a simple comparison without any other evidence of correlation that states OSM data quality has a strong correlation with income per capita. The author can discuss this further in the discussion section. References should be added that are more geared towards the findings of this research. This output catches the reader's attention, but since it is not referenced and discussed further, it is seen as an ambiguous statement.
Author Response
Dear Reviewer 3,
We would like to thank you for your valuable comments and recommendations. We tried to make the revisions as carefully as we could, and we used all your comments and recommendations. Concerning the pieces of advice, we modified parts of the manuscript and marked them in red. Below, we presented, in brief, the replay to your comments.
Please see the attachment.

Reviewer 4 Report
- the article seems to be oriented only to selected number of county, but it would be interesting to see results of analysis for all counties in Poland.
- Only on few places in Introduction part of the paper authors deals with previous work in the field. I suggest to authors to add custom section "Literature review" or "Related work" where they will sublime current state of the art, and to emphasize difference to their analysis.
- Authors should consider restructuring of whole Section 2. For example, the name of subsection 2.1.1 is completely related to the structure of the OSM not to the structure of the paper which can confuse the reader. Similarly, section 2.1.2. deals with methods of obtaining data in OSM, not with obtaining data related to paper, which is confusing.
- Authors to consider to extract subsection 2.1 into separate section and to rename it.
- Sentence "In research [18] explain this incompatibility by the impossibility of reconciling the idea of "wiki-style" work with the commonly accepted standards in the field of geographic information." should be reformulated. Beginning of the sentence is not clear enough.
- Authors to consider to extract section 2.2 into separate section. This section could be named "Materials and methods", not the previous one.
- Also, I recommend authors to create one custom section where methods of research described in the paper are clearly presented in order that reader can easily follow up. Maybe some figure of proposed methodology or something similar.
- Table 9. and is partially on page 19, and mostly on page 20. Place whole table on one page.
- Similarly for table 11.
- The heading of subsection 4.4 starts on page 23 and text belonging to it on next page. Please correct this.
- I miss the future recommendations for research at the ending section of the paper. Authors should discuss this and give their opinion on how geospatial data quality should be maintained and improved.
Author Response
Dear Reviewer 4,
We would like to thank you for your valuable comments and recommendations. We tried to make the revisions as carefully as we could, and we used all your comments and recommendations. Concerning the pieces of advice, we modified parts of the manuscript and marked them in red. Below, we presented, in brief, the replay to your comments.
Please see the attachment.

Round 2
Reviewer 2 Report
The updates on the manuscript have for sure improved readability and clearness of their content. Anyway, in my opinion, a few elements have to be still clarified before accepting its publication:
- Regarding the semantic accuracy, in section 3.2.5 it is still not clear which criteria of accuracy have been used. What is the meaning of the sentence "F – number of objects with the correct name in OSM dataset". How the authors have defined whether a name is correct or not? In comparison to what other information? Could it be that the number of objects with additional tags to the main one, such as the name and others describing the additional type of mapped element, has been taken into account instead? This has to be better explained here and discussed clearly in 5.5.
- The issue of imports in the coverage or quality of OSM has been just mentioned ("Some of the people editing OSM import BDOT10k data on their own, but there is no unambiguous information available about the number of introduced objects.") and not tackled or discussed. It seems that several buildings imports were performed in Poland: https://wiki.openstreetmap.org/wiki/Pl:Importy_oficjalnych_danych_pa%C5%84stwowych Has this been considered?
- regarding my previous comment on lines 194-200 ("in describing the novelty of the research, the focus of the authors is about analysis of land cover, which, per se, is not a novelty in OSM quality assessment. The are already in OSM studies on data quality in relation to land use and also recently about integration of authoritative data at national level and not only at city or agglomeration (which could be cited here). The novelty seems to me to have these elements analysed also in relation to SDGs." the authors answered "This valuable suggestion has been incorporated in the text", but it is not clear to me where this happened.
- in the discussion section, paragraphs 5.1, 5.2, 5.3, 5.4 still seem to be a summary of the results, not actual discussion of them; it is not clear if the text from line 711 are referring to the current 5.5 section. If these considerations are including more general elements, they should be grouped under a different section.
Some minor notes:
- line 453-454: "quantitative analysed" > "quantitative analyses"
- line 791: "Open-StreetMap" > "OpenStreetMap"
- lines 790-798: please add relevant bibliographic references to this section
- lines 815-816: "The value of coverage of OSM linear and surface objects with the reference base was in the counties Słupski 55.4%, Sanocki 55.2%, Ostrowski 51%". The sentence seems to be uncomplete
Reviewer 3 Report
-
Reviewer 4 Report
Accept in current form
